



# Trends in the annual snow melt-out day over the French Alps and the Pyrenees from 38 years of high resolution satellite data (1986-2023)

Zacharie Barrou Dumont[1], Simon Gascoin[1], Jordi Inglada[1], Andreas Dietz[2], Jonas Köhler[2], Matthieu Lafaysse[3], Diego Monteiro[3], Carlo Carmagnola[3], Arthur Bayle[4], Jean-Pierre Dedieu[5], Olivier Hagolle[1], and Philippe Choler[4]

[1]Centre d'Etudes Spatiales de la Biosphère, CESBIO, CNES/CNRS/IRD/UT3 Paul Sabatier, Toulouse, France.
[2] German Remote Sensing Data Center (DFD), German Aerospace Center (DLR), Muenchener Strasse 20, D-82234 Wessling, Germany.
[3]Univ. Grenoble Alpes, Université de Toulouse, Météo-France, CNRS, CNRM, Centre d'Etude de la Neige, Grenoble, France
[4]Univ. Grenoble Alpes, Univ. Savoie Mont Blanc, CNRS, Laboratoire d'Ecologie Alpine (LECA), 38000 Grenoble, France.
[5] Institut of Environmental Geosciences (IGE), Université Grenoble Alpes/CNRS/Grenoble INP/ INRAE / IRD, France.

**Correspondence:** Zacharie Barrou Dumont (zachariebd@hotmail.com) and Simon Gascoin (simon.gascoin@univ-tlse3.fr)

**Abstract.** Information on the spatial-temporal variability of the seasonal snow cover duration over long time periods is critical to study the response of mountain ecosystems to climate change. However, this information is often lacking due to the sparse distribution of in situ observations or the lack of adequate remote sensing products. Here, we combined snow cover data from ten different optical platforms including SPOT 1-5, Landsat 5-8 and Sentinel-2A&B to build a time series of the annual

snow melt out day (SMOD, i.e. the first day of no snow cover) at $20\,\mathrm{m}$ resolution across the French Alps and the Pyrenees ($43 \times 10^3\,\mathrm{km}^2$). We evaluated the pixel-wise accuracy of the computed SMOD using in situ snow measurements at 344 stations. We found that the residuals are unbiased (median error of 1 day) despite a dispersion (RMSE of 28 days), which suggests that this dataset can be used to study SMOD trends after spatial aggregation. We found an average reduction of 20.4 days (5.51 days per decade) over the French Alps and of 14.9 days (4.04 day per decade) over the Pyrenees over the period 1986-2023. The

SMOD reduction is robust and significant in most part of the French Alps and can reach one month above 3000 m. The trends are less consistent and more spatially variable in the Pyrenees. This dataset is available for future studies of mountain ecosystems changes and is updated every year using Sentinel-2 data.

## 1 Introduction

In mountain regions, hydrological and ecological processes are strongly influenced by the seasonal snow cover. As a result,

ongoing snow cover changes due to global warming threaten the sustainability of numerous ecosystem services (Hock et al., 2019; Adler et al., 2022). Specifically, the annual snow melt out day (SMOD, also named the snow cover melting day, first day of no snow cover or first no-snow day) modulates the onset of the vegetation growing season and therefore has a profound



impact on soil processes, vegetation phenology and productivity (Alonso-González et al., 2024; Choler, 2015; Francon et al., 2023; Jonas et al., 2008; Revuelto et al., 2022; Edwards et al., 2007; Choler et al., 2024).

The Copernicus Climate Change Service report on European State of the Climate highlights that Europe is the fastest warming continent in the world, with a warming occurring at a rate twice that of the rest of the world (Copernicus Climate Change Service (C3S), 2024). In the French Alps from 1959 to 2010, the near-surface air temperature has increased at a rate of 0.25-0.4 +/- 0.2 degrees per decade (Beaumet et al., 2021), consistent with rates reported in the rest of the European Alps (Hock et al., 2019; Rottler et al., 2019; Scherrer, 2020; Monteiro and Morin, 2023). In the Pyrenees, the air temperature increased by
0.2°C per decade between 1959 and 2010 (OPCC-CTP, 2018). Previous studies have also identified long-term changes in the seasonal snow in European mountainous areas (Notarnicola, 2022), including the Pyrenees (López-Moreno et al., 2020; Pons et al., 2010; Vidaller et al., 2021) and the Alps (Durand et al., 2009; Marty et al., 2017; Matiu et al., 2021; Scherrer et al., 2004; Schöner et al., 2019; Valt and Cianfarra, 2010; Dedieu et al., 2014). In the Mont-Blanc massif, the highest peak of the European Alps, SMOD decreased at all elevations between the two decades 1965-1975 and 2005-2015 (CREA, 2024). However, little is
known about the SMOD trends at larger scale in the European mountains, limiting our ability to understand and anticipate the response of mountain ecosystems to climate change. This work aims to fill this gap in the French Alps and the Pyrenees.

To achieve this objective, we defined three requirements: (i) a spatial resolution lower than 100 m, (ii) a temporal depth exceeding 30 years, and (iii) an effective revisit frequency better than one month. The first requirement comes from the complex topography of mountainous regions. In particular, the snow cover sensitivity to climate warming varies with the terrain slope
and aspect (López-Moreno et al., 2014) and the spatial variability of mountain snow depth is typically within a range of less than 100 m (Blöschl, 1999; Mendoza et al., 2020; Trujillo et al., 2007). Hence, resolutions lower than 100 m are needed to decipher the influence of snow on alpine vegetation (Dedieu et al., 2016). The second requirement aims to limit the impact of the natural climate variability such as the North Atlantic Ocean oscillation, which extends to decadal scales in western Europe (Hurrell, 1995). In the European Alps and the Pyrenees, the sign and magnitude of snow cover trends becomes unstable if periods
shorter than 30 years are considered (López-Moreno et al., 2020; Monteiro and Morin, 2023). In general, at least 30 years is recommended to enable the analysis of climate-driven snow trends (Bormann et al., 2018). The third requirement is motivated by the need to identify the SMOD with a sufficient accuracy to analyse its trend, whose magnitude could be of the order of 1 week to 1 month over the past three decades (e.g. Hüsler et al., 2014; López-Moreno et al., 2020; Durand et al., 2008; Matiu et al., 2021). The accuracy of the SMOD is directly linked to the density of snow observations throughout the year (Hüsler
et al., 2014).

We reviewed the different data sources that could be used to fulfill these requirements: in situ data, numerical snowpack modeling and remote sensing. in situ measurements face significant limitations due to the generally scarce number of long-term high-elevation sites and their uneven distribution (López-Moreno et al., 2020; Matiu et al., 2021). Stations above 2500 m are rare, representing 5% of the stations used by Monteiro and Morin (2023) and 3% of the stations used by López-Moreno
et al. (2020) in the French Pyrenees. This hampers the interpolation of in situ trends at regional scales (Rohrer et al., 2013). Snowpack modeling can provide spatially continuous snow cover information. However, the temporal heterogeneity of the





available meteorological data used for the atmospheric forcings can add biases to the long-term trends especially in high elevation regions due the scarcity of meteorological stations (Vernay et al., 2022).

Remote sensing is an effective tool to study the snow cover evolution over large regions. AVHRR data (1 km resolution) were used to determine snow cover trends since 1985 in the Alps (Hüsler et al., 2014). The MODIS instrument, active since 2000, allowed the production of a 500 m resolution dataset of daily global snow cover parameters (Dietz et al., 2015) and was used to show a negative snow cover duration trend of 17 days per decade across the Alps (Fugazza et al., 2021). The American Landsat program provides the opportunity to map the snow-covered area at higher resolution (60-30 m) since 1972, with each of the satellites from Landsat 1 to Landsat 9 capturing freely available images of continental surfaces with a revisit of 16 days. The program kept two Landsat satellites in activity at the same time, improving the revisit to 8 days (Loveland and Irons, 2016). This combination of available temporal depth and decametric spatial resolution unlocked new analyses of the snow cover dynamics in mountain regions (Choler et al., 2024; Bayle et al., 2023; Carlson et al., 2017; Margulis et al., 2016; Hu et al., 2020; Rumpf et al., 2022; Koehler et al., 2022a, b). Since 2017, our capability to characterize the snow cover evolution in mountains has further progressed with the Sentinel-2 mission, which offers a unique combination of systematic global coverage of land surfaces, 5-day revisit time under consistent viewing conditions, high spatial resolution (10, 20, and 60 m) and multi-spectral observations (Gascoin et al., 2019). Therefore, satellite remote sensing appears as a promising and relevant method to address the objective of this work. However, most satellite missions with freely available data fulfill only one or two of the requirements defined above. Despite their daily revisit, MODIS and AVHRR spatial resolutions are not sufficient. MODIS and Sentinel-2 periods of record are still too short. Landsat mission revisit times may be theoretically sufficient but are not guaranteed due to technical constraints (Ju and Roy, 2008; Zhang et al., 2022). Accounting for cloud cover, the effective revisit of Landsat is approximately one observation per month or less which hinders applications in mountain ecosystems (Bayle et al., 2024b).

In 2015, the French Space Agency (Centre National d'Études Spatiales, CNES) started to open the archive of SPOT images through the SPOT World Heritage project (SWH). The SPOT (Satellites Pour l'Observation de la Terre) program consists of five satellites, which observed the Earth with 3 to 4 spectral bands with spatial resolutions of 10 and 20 m. SWH led to the release of nearly 20 million SPOT 1-5 products from 1986 to 2015. Since SPOT acquired images on demand, determining the average revisit frequency is challenging. Nevertheless, as acquisitions over west Europe were requested more often, we found that SWH significantly increases the number of available observations from Landsat only over the French Alps and Pyrenees (Barrou Dumont et al., 2023). In these regions, SWH offers the opportunity to enhance the temporal revisit of high resolution multispectral imagery since the 1980's, thereby helping us to fulfill the above requirements. The low radiometric depth and the absence of a Short Wave InfraRed band (SWIR) in SPOT images pose challenges for snow cover and cloud classification. However, we showed that this issue can be efficiently addressed using an image emulation approach to train a deep-learning algorithm (Barrou Dumont et al., 2024b).

In this article, we merged the time series of SPOT, Landsat and Sentinel-2 snow cover products to compute SMOD trends over the period 1986-2023 covering a domain of $43 \times 10^3 \ \mathrm{km}^2$ including the French Alps and the Pyrenees at 20 m resolution ($1.1 \times 10^8$ pixels). We evaluated this new dataset using in situ SMOD observations in both mountain ranges. The combination





of high spatial resolution and temporal depth of this dataset allowed us to analyze spatial variability of SMOD trend at fine spatial scales. The analysis was stratified by region of approximately 1000 km$^2$, and within each massif by topographic class (Fig. 1). These regions called "massifs" are relatively homogeneous with respect to their principal climatological characteristics

at a given elevation, slope, and aspect (Durand et al., 1999).

## 2   Data

### 2.1   Satellite products

We describe the satellite datasets following the remote sensing nomenclature which separates products by processing level. Level 1C refers to orthorectified top-of-atmosphere reflectances. Level 2B refers to a labelled image of the surface properties

at the time of the acquisition, in this case three classes: snow, no-snow, cloud. The three collections of satellite data used in this work are SWH (SPOT 1-5), DLR-Landsat (Landsat 5-8) and Theia (Sentinel-2A&B and Landsat 8).

#### 2.1.1   SWH

Each SPOT had two identical instruments. The first generation SPOT 1-3 were equipped with twin High Resolution in the Visible (HRV) instruments with green, red and NIR bands at 20 m spatial resolution. SPOT 4 was equipped with twin High

Resolution in the Visible and Infrared (HRVIR) instruments which had the same geometric multispectral imaging characteristics as the HRV but with the addition of a 20 m resolution SWIR band. The SPOT 5 High Geometrical Resolution (HRG) twin instruments had the same multispectral characteristics as HRVIR except for an improved spatial resolution of 10 m for the three visible bands. The Theia data and services hub (https://www.theia-land.fr/product/SPOT-world-heritage-fr/) provides a subset of $130\,514$ SPOT 1-5 $60 \times 60$ km$^2$ images that were processed to level 1C mostly over the French territories and Africa.

$22\,868$ of those level 1C images cover the French Alps and Pyrenees from 1986 to 2015 and were used in this study. We processed them to level 2B using an algorithm which was designed to cope with the low radiometric quality of SPOT images and the lack of shortwave infrared band in SPOT 1-3 (Barrou Dumont and Gascoin, 2021). This algorithm was marginally adjusted for this study to reduce an overestimation of the cloud mask. These adjustments are described in appendix A.

#### 2.1.2   DLR-Landsat

We used level 2B products generated from Landsat 5 to 8 images over the Alps by Koehler et al. (2022a, b). Their algorithm was used to generate the same products specifically over the Pyrenees for this study. We used these products for the period 1986-2015 (up to 2017 for Landsat 7). On average, those Landsat-derived 2B products achieved an overall accuracy of 87.5 % for the Landsat 5 TM sensor, 95.5% for the Landsat 7 ETM+ sensor, and 95.5 % for the Landsat 8 OLI sensor (Koehler et al., 2022a).



### 2.1.3 Theia

For the period 2016-2023, we used the level 2B products derived from Sentinel-2A&B and Landsat 8 that are routinely distributed by Theia (Gascoin et al., 2019). These products are available over the study area since 2015 if the cloud fraction in the image is below 90%. Landsat 8 processing was discontinued in August 2021. After that date, the only source of data is Sentinel-2, however this only degrades the revisit time from 4 to 5 days. The SCA detection was evaluated in the French Alps and Pyrenees and had an accuracy of 94 %(Gascoin et al., 2019). A more comprehensive evaluation was conducted at pan European scale and yielded comparable results (Barrou Dumont et al., 2021).

### 2.2 Auxiliary data

The elevation of analyzed pixels was sourced from the Copernicus 30 m resolution digital elevation model GLO-30 (ESA and Airbus, 2022). The forest cover was derived from the Copernicus 20 m resolution tree cover density (TCD) for 2015 (EEA, 2018). The glacier areas were obtained from the Randolph Glacier Inventory (RGI) version 6.0 which provides the glacier outlines for 2010 (RGI Consortium, 2017). The water mask was derived from the Copernicus EU-Hydro river network database for 2006-2012 (EEA, 2020).

### 2.3 in situ data

We used time series of snow depth at ground stations from the Météo-France network. Each time series was obtained by assimilating in situ snow depth observations in a model run forced by the SAFRAN–SURFEX/ISBA–Crocus–MEPRA (S2M) reanalysis (Vernay et al., 2022). The assimilation method is the direct insertion, which means that the reconstructed snow depth is equal to the observation when available. This method was developed to fill gaps in snow depth time series measurements due to sensor failures or absence of the human observer (in the case of manual measurements). This dataset was previously used in the Pyrenees (López-Moreno et al., 2020). Station coordinates were not recorded with a consistent precision (between $10^{-1}$ and $10^{-6}$ degrees). Since the size of a 20 m pixel corresponds to the $10^{-4}$ degrees order, only stations with position precision of $10^{-4}$ and below were used. We excluded stations located in pixels with a tree cover density greater than 50% as done for the satellite data processing (Sect. 3.2). Some stations became active later in the study period, and other stations ceased activity earlier in the period but, overall, the number of active stations increased over the years (Fig. 2).



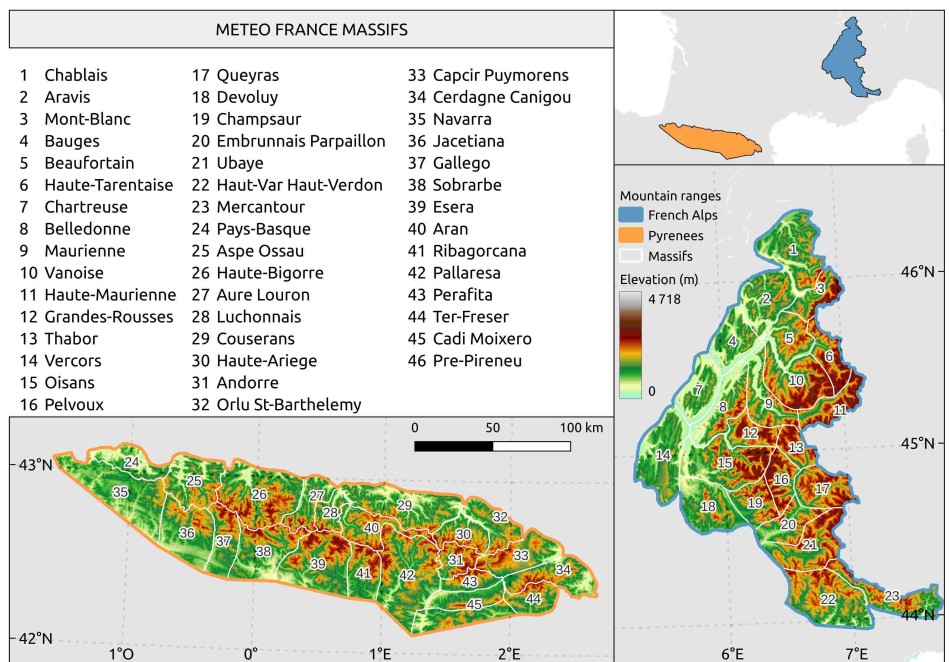

**Figure 1.** Pyrenees and French Alps divided into 23 massifs each.

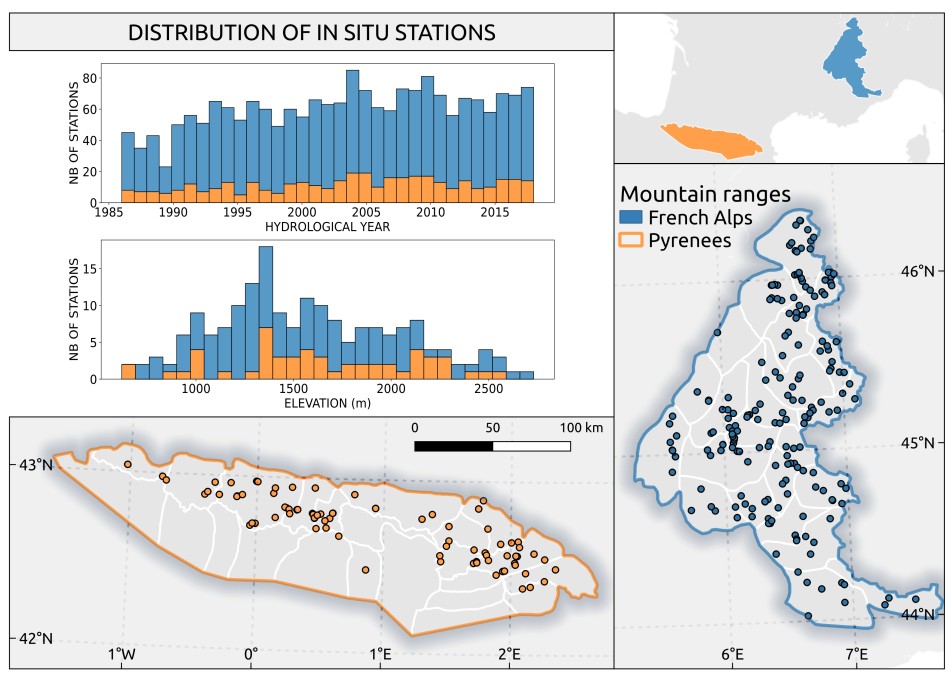

**Figure 2.** in situ snow depth stations distribution.



## 3 Method

This section presents the method to generate 20 m resolution SMOD time series and associated trends. The SMOD were determined by hydrological year (HY) starting on 01 September. For example, the period between 01 September 2000 and 31 August 2001 referred to the HY 2000, or 2000/2001. The trends were computed from the combination of three above collections of level 2B snow products (SWH, DLR-Landsat and Theia). We merged the SWH and DLR-Landsat datasets into a single dataset covering the HY 1986 to 2014 and which we referred to as SWHLX 3. Since SWH and DLR-Landsat datasets

were generated from different approaches, we also evaluated their agreement using products that were generated from images acquired within the same day. Then, we concatenated the SWHLX and Theia datasets (Fig. 3). For the HY 2015 and 2016, the DLR-Landsat products (Landsat 7) were added to the Theia dataset to compensate the absence of Sentinel-2B (launched in 2017).

### 3.1 Evaluation of SWH and DLR-Landsat agreement

The DLR-Landsat dataset initially only covered the French Alps with 2916 products from 5 Landsat tiles (acquisition path/row combinations 195-196/28 and 194-196/29), each covering an approximate area of $173 \times 183$ km$^2$. It was extended for this work with 3079 Landsat 5-9 images from 5 Landsat tiles (acquisition path/row combinations 198-200/30 and 197-198/31) covering the Pyrenees, reaching a total of 5995 Landsat 2B products. The Pyrenees images were selected and the 2B products generated following the same methodology as Koehler et al. (2022a, b). Landsat 4 images were not included in the study as their number

was negligible in the area and period of study.

To assess the differences and similarities between the SWH and DLR-Landsat datasets, pairs of overlapping SPOT and Landsat 2B products were assembled from same day acquisitions and reprojected to 30 m resolution (using nearest neighbor resampling). A comparison of the two datasets was firstly done qualitatively with a visual comparison of the 2B products between the SPOT instruments HRV, HRVIR and HGR and the Landsat instruments TM, ETM+ and OLI in both cloudy and

cloud-free contexts. A quantitative assessment was also conducted by computing a contingency matrix of the labels, excluding forest pixels (TCD > 50%). The comparison was restricted to the snow and no-snow labels, excluding cloud pixels, because cloud cover can change in the time interval between two same-day acquisitions. Landsat 2B products have a shadowed areas and

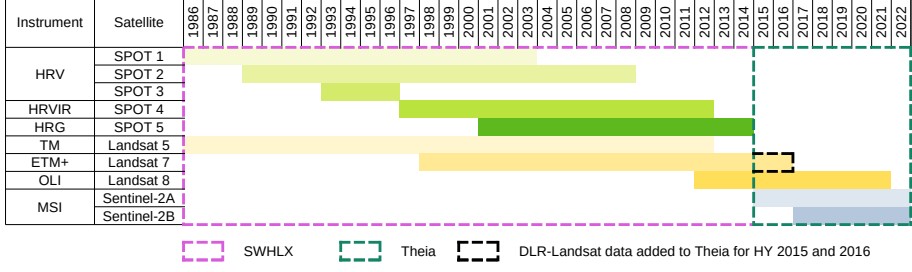

**Figure 3.** Satellite products distribution in the time series across the hydrological years.





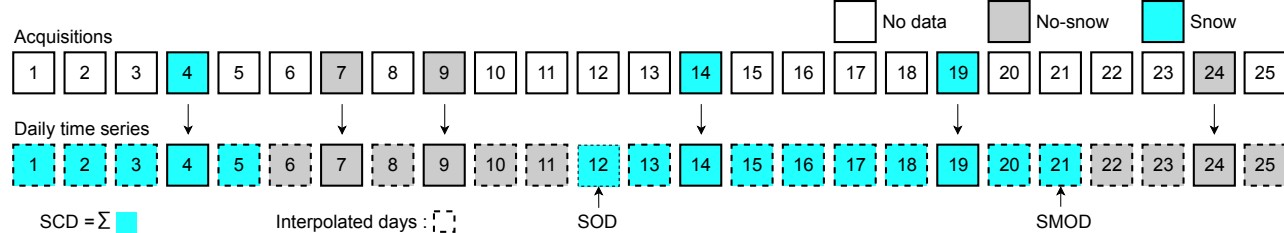

**Figure 4.** 25 days hydrological period of snow, no-snow and no-data days representing how snow periods are interpolated. The longest Snow Cover Duration (SCD) starts with the Snow Onset Day (SOD) on day 12 and ends with the Snow Melt-Out Day (SMOD) on day 21

water bodies label, which was merged with the cloud label as no-data values. From the contingency matrix between SWH and DLR-Landsat data, we derived the F1, recall and precision scores of SWH (see appendix B). The use of those metrics typically
implies that a predicted dataset is compared to a reference dataset but, in this case, they served to study the similarities between both datasets.

## 3.2   Calculation of the snow melt-out day

We defined the SMOD as the last day of the longest uninterrupted snow-covered period in the hydrological year (Monteiro and Morin, 2023). All SPOT and Landsat products were reprojected using a nearest-neighbor interpolation and cropped to the
same reference system as Sentinel-2 products, i.e. a 20 m resolution grid in the Universal Transverse Mercator organized by tile of 110 km by 110 km called the Military Grid Reference System. To obtain a daily time series of snow and no-snow labels from the 2B products time series, we used a nearest-neighbour interpolation between snow (1) or no-snow (0) observations in the time dimension (Fig. 4). This process also filled missing data due to cloud cover or the failure of the scan line corrector in Landsat 7 imagery. For the Theia dataset, we interpolated daily values near the beginning and the end of each HY by using 2B
Theia products found within a margin of 15 days outside the HY. We observed that this method was insufficient with SWHLX in some areas with a short no-snow season due to its lower revisit time. Hence, we assumed that persistent snow cover is negligible outside the glacier areas as defined by the RGI and assigned all non-glacier pixels the no-snow label on the day before and the day after each HY to improve the interpolation near the beginnning and the end of the HY in case of long period of without observations. For cases where two level 2B products from different sensors were available on the same day, the
products were merged to a single raster by giving priority to (i) clear sky observations (ii) the pixel value from the highest resolution sensor. For example, a SPOT snow pixel (20 m) had priority over a Landsat no-snow pixel (30 m), but a Landsat no-snow pixel had priority over a SPOT cloud pixel. Finally, we marked as no-data the SMOD pixels whose tree cover density was greater than 50%, or which are located in the glacier mask or in the water mask.



### 3.3 Evaluation of snow melt-out days

We evaluated the satellite-derived SMOD from SWHLX and Theia with in situ SMODs (Sect. 2.3). To compute the in situ SMOD from the reconstructed in situ snow depth time series (Sect. 2.3), we used a snow depth threshold $HS_0$ to separate the daily snow depths into binary snow/no-snow labels, i.e. a snow label was assigned if the snow depth was greater or equal to $HS_0$. We used a $HS_0$ value of 1 cm as it was already identified as the best threshold for Sentinel-2 snow products (Barrou Dumont et al., 2021). We tested the sensitivity of the results to $HS_0$ by increments of 1 cm and found no significant differences for

values below 5 cm (not shown here). Then, we computed the SMOD for each hydrological year from the same definition, i.e. the last day of the longest uninterrupted snow period. We excluded every SMOD value which coincided with a missing value in the original observed time series, since it was reconstructed by the model at that time step. In this way, we only evaluated the satellite SMOD with observed values, but we still took advantage of the entire reconstructed time series to identify the longest uninterrupted snow period. The agreement between the in situ and satellite SMOD was evaluated using the statistical

distribution of the residuals for every hydrological year at each station $\Delta$SMOD (Eq. 1)

$$\Delta \mathrm{SMOD} = \mathrm{SMOD}_{\mathrm{year,station}}^{\mathrm{satellite}} - \mathrm{SMOD}_{\mathrm{year,station}}^{\mathrm{insitu}} \tag{1}$$

The SMOD accuracy is expected to increase with the number of clear-sky observations, as a higher number of clear-sky observations reduces the importance of the interpolation in the SMOD calculation. Therefore, we sought to establish a relationship between NOBS and the SMOD errors. However, our in situ dataset did not sample the full distribution of topography in the

study region. Thus, we generated an additional reference dataset from the Theia products, taking advantage of the high revisit of combined Sentinel-2 and Landsat 8 acquisitions to produce a spatially-distributed reference SMOD dataset. We selected all Theia level 2B products of tile 31TCH (Pyrenees) for HY 2017. We randomly sampled a subset of these products, following the seasonal probability of SPOT and Landsat acquisitions during the period 1986-2014. This probability was obtained by normalizing the seasonal distribution of SWHLX data from HY 1986 to HY 2015 (Tab. 1). From this undersampled time

series of Theia products, we re-computed the SMOD and NOBS. The process was repeated by incrementally increasing the sampling size from 10 images to the number of available Sentinel-2 images minus 1. At each increment, the random sampling was repeated five times. This generated dataset allowed us to study the relationship between the sampled $\Delta$SMOD and NOBS value. To characterize the SMOD errors, we used the Median Absolute Error (MAE), the Root Mean Squared Error RMSE and the Interquartile Range (IQR) (appendix B).

We also evaluated the temporal stability of the SMOD error, because a time-dependant error could create spurious trends (Bayle et al., 2024b). Hence, we analyzed the temporal evolution of $\Delta$SMODs at two stations where SMOD values are available over the entire study period while respecting the minimum of observations that we defined as a threshold to compute trends (see Sect. 3.4).





| Autumn | Winter | Spring | Summer |
|---|---|---|---|
| Sept - Nov | Dec - Feb | Mar - May | Jun - Aug |
| 0.255 | 0.208 | 0.240 | 0.297 |

**Table 1.** Seasonal weights used to sample Theia products to emulate SWHLX acquisitions over tile 31TCH

### 3.4 Snow melt-out day trends

SMOD trends were computed using the Mann-Kendall (MK) test on the combined time series of SWHLX and Theia. The Mann–Kendall (MK) test is a method to detect consistently increasing or decreasing trends in temporal series (Kendall, 1948; Mann, 1945) and is often used in snow hydrology, including SMOD trend analysis (Klein et al., 2016; Nedelcev and Jenicek, 2021; Wang et al., 2016; López-Moreno et al., 2020). We used the Python implementation *pymannkendall* (Hussain and Mahmud, 2019) to compute the MK test and the slope of the trend with the Theil-Sen method (Sen, 1968). We considered that

a trend was statistically significant if the p-value was below 0.05. To evaluate the distribution of trends by elevation, we discretized the digital elevation model with the same 300 m elevation bands of as the SAFRAN system (Durand et al., 1999). We restricted this analysis to elevation bands 1500 m $\pm$ 150 and above to focus on areas of seasonal snow cover. As a reference, in the Pyrenees, there is snow on the ground at least 50 % of the time between December and April at elevations above 1600 m (Gascoin et al., 2015). For every triplet of year, massif, and elevation band, we calculated the median of all the corresponding

SMOD pixel values (no glacier, TCD $\leq$ 50). We excluded the triplets for which less than 1000 valid SMOD values were available. For every combination of massif and elevation band, a trend was obtained by calculating the Sen slope from the SMOD time series.

Trends were also calculated by classes of topographic aspect and Diurnal Anisotropic Heat (DAH). DAH represents the distribution of the heating of the surface by the sun. It has been used to study snow spatial variability during ablation season

(Cristea et al., 2017) and plant growth responses to changes in snow cover in above-treeline elevations (Choler, 2023). It can be approximated by equation 2 (Böhner and Antonić, 2009) where $a$ is the aspect of the slope, $\beta$ the slope angle, and $\alpha_{max}$ is the aspect with the maximum total heat surplus after a day/night cycle.

$$\mathrm{DAH} = \cos(\alpha_{\mathrm{max}} - \mathrm{a}) \times \arctan \beta \qquad (2)$$

We used $\alpha_{\mathrm{max}} = 212°$ (a direction between south and south-west) following Bayle et al. (2024a).

Since the accuracy of SMOD is expected to decrease when the number of observations is low, we only considered the cases when NOBS was above a threshold. This minimum NOBS threshold ($\mathrm{NOBS}_{\mathrm{min}}$) was applied in the SMOD pixels selection process before calculating the SMOD medians. However, a high $\mathrm{NOBS}_{\mathrm{min}}$ threshold might prevent the calculation of SMOD medians for some combinations of massif and elevation band, especially in the 1986 and 1987 years when available acquisitions were lower. Hence, we assessed the sensitivity of SMOD medians and their availability to $\mathrm{NOBS}_{\mathrm{min}}$.



Another source of uncertainty is the interannual variability which can create spurious trends if the period of trend calculation is too short with respect to the natural variability. To test the robustness of our SMOD trends to the study period, we computed a trend matrix of every possible time period of 20 HY or more between 1986 and 2022 for every combination of massif and elevation band. The trend of a period was calculated only if the SMOD median values of the first and last HY of the period were available and if at least 95% of the period's years were available (19 years for a period of 20 years, 35 years for a period

of 37 years).

## 4    Results

The first part of this section presents the assessment of historical SWH 2B products in comparison to the DLR-Landsat data. The second part presents an evaluation of SMODs obtained from both SWHLX and Theia against in situ data, as well as a study of their uncertainties. The third part answers the second objective of this work and leverages the high spatial resolution

of the data to establish SMOD trends for HY 1986 to HY 2022 stratified in respect to the massifs delimited by Météo-France and with respect to topographic classes.

### 4.1    Evaluation of SWH and DLR-Landsat agreement

We assembled 1340 pairs of Landsat (DLR-Landsat) and SPOT (SWH) products acquired on the same day. There were more snow pixels and fewer cloud pixels from SWH than from DLR-Landsat (Tab. 2), and the differences were larger in the Alps

than in the Pyrenees. When cross-masking cloud labels from both datasets, both products had a high agreement with each other, with a snow recall of 0.99 and a snow precision of 0.98 for a F1 score of 0.98. Visual comparisons can be found in appendix (Fig. C1, C2, C3). They illustrate the overall good agreement between both datasets despite the different methods to generate them, as well as the differences in the classifications. Border areas between snow-covered and no-snow regions contained ambiguous pixels harder to classify correctly (Fig. C1). The DLR-Landsat method hid those pixels under a cloud

label, avoiding the risk of generating false snow positives. Inversely, SWH 2B products classified those pixels as either snow or no-snow. Other noticeable differences between DLR-Landsat and SWH were that SWH overestimated the size of existing clouds (Fig. C2) and that shadowed snow areas were misclassified as clouds with the DLR-Landsat approach (Fig. C3).

| Mountain range | French Alps | | Pyrenees | |
|---|---|---|---|---|
| Nb of pixels / Nb of pairs | 3.3e8 / 726 | | 4.3e8 / 614 | |
| Dataset | SWH | DLR-Landsat | SWH | DLR-Landsat |
| % of snow labels | 13 | 10.9 | 8.8 | 7.3 |
| % of cloud labels | 28 | 31.9 | 26.7 | 27.2 |

**Table 2.** Distribution of snow/cloud labels among the SWH and DLR-Landsat pairs.





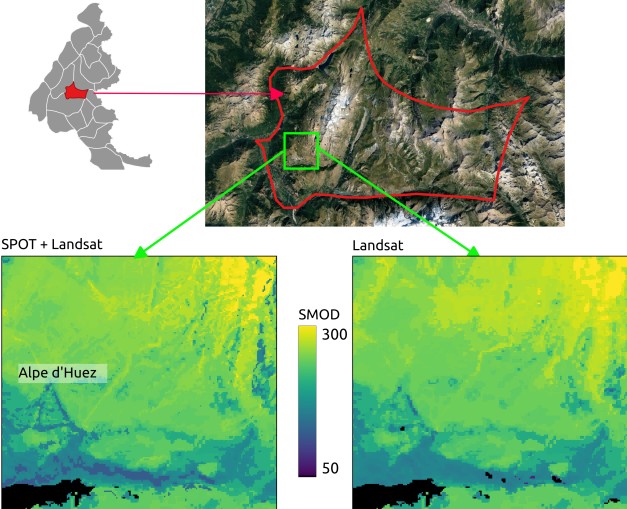

**Figure 5.** Alpe d'Huez, hydrological year 1997: comparison between the 20 m resolution SMOD maps from Landsat (left) and from Landsat + SPOT (right).

A benefit of using SWH is that SMOD images can be computed at 20 m instead of 30 m when using Landsat only. The enhanced spatial temporal resolution of a combined SWH-Landsat dataset results in SMOD image with more spatial details than a SMOD image from Landsat only (Fig. 5).

### 4.2 Evaluation of snow melt-out days

Our in situ data spanned HY 1986 to 2018. With $HS_0 = 1$ and for the Theia dataset, we found an MAE of 9 days, an RMSE of 24 days and a median $\Delta$SMOD of -1 day (Fig. 6). Combining Theia with SWHLX gave an MAE of 11 days, an RMSE of 28 days and a median $\Delta$SMOD of 0 day. Both SWHLX and Theia performed better at higher SMOD corresponding to June and July (i.e., above 275 days).

Both in situ and satellite-based SMODs tended toward 200 (19-20 March) at lower elevations (1200 $\pm$ 150 m) and up to 280 (7-8 June) at 2400 $\pm$ 150 m (Fig. 7). We observed no significant bias toward an overestimation or underestimation with satellite-based SMODs in the Alps, but we observed an overestimation in the Pyrenees for every available elevation band excepted for 2400 m. This calls for caution when interpreting trends in the Pyrenees. The better performance in the Alps contributed to the overall results due to the lower number of in situ data in the Pyrenees and their lower elevation. From both stations which provide a SMOD reference value over the full study period, we did not find a clear trend in the evolution of the bias over time (Fig. C4)

The synthetic dataset derived from Theia products allowed sampling a larger amount (> 10 000) of data (Fig. 8). We found an error variance significantly higher than 0 for NOBS $\leq$ 26 and a bias for NOBS $\leq$ 11 which starts increasing for NOBS $\leq$





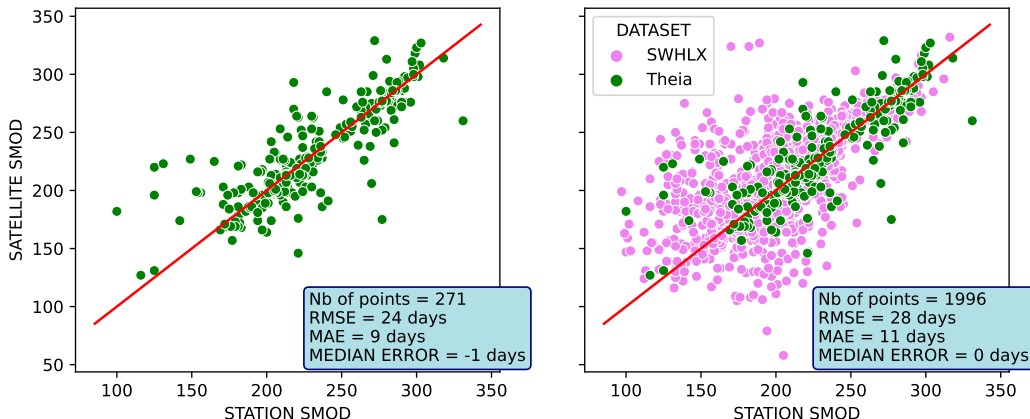

**Figure 6.** Scatterplots between satellite-based SMODs and station SMODs at $HS_0 = 1$ for Theia between the HY 2015 and 2018 (left) and for the combination of Theia and SWHLX between the HY 1986 and 2018 (right).

7. Excluding outliers, $\Delta$SMOD was below 30 d for NOBS $\geq$ 19. We observed a negative correlation between NOBS and the different error metrics for NOBS $\leq$ 26 (Fig. 8). Beyond NOBS = 26 the error metrics were less sensitive to the NOBS value.

### 4.3 Snow melt-out day trends

Based on the previous analysis, we selected $NOBS_{min}$=10 as the selection threshold before agregating SMOD values by regions and to compute the SMOD trends. We found that higher NOBS values led to a significant reduction of available combinations of HY, massif and elevation bands from which a SMOD median could be calculated (Fig. C5 in appendix). Despite the large error that can be obtained with NOBS = 10 at the pixel level, the median values by massif and elevation bands were similar to the ones that can be obtained with higher $NOBS_{min}$ (Fig. C6 in appendix). For those reasons, an $NOBS_{min}$ of 10 was chosen to report the results below, but we repeated the analysis with higher values and found similar results, albeit with more gaps.

For the HY 1986 to 2022, all statistically significant trends were negative. In the French Alps, non-significant trends were either negative or under $\pm$1 day per decade (Fig. 9). Significant negative trends were mostly situated in the northern massifs at 1500 m, in the center east at 1800 m, and in the center south at 2100 m. Trends became generally weaker at 2400 m and stronger again at 2700 m and above. At 2400 m and above, the Mont-Blanc massif had a consistently significant decreasing trend which reached -14 days per decade at 3000 m.

In the Pyrenees, all statistically significant trends were also negative and mostly found in the eastern half (Fig. 10). The Pre-Pireneu and Cadi Moixero massifs were not represented because they were not sufficiently covered by the Sentinel-2 tiles. Negative trends were less pronounced as the Alps, and non-significant positive trends were also observed below the 2100 m elevation band. Most of the data at 1500 m was missing in the Spanish side due to the lack of SPOT 1C images further inside Spanish territory. Stratification of the analysis in respect to aspect and DAH revealed significant trends (Fig. C7, C8, C9, C10),



**Figure 7.** Normalized histogram of the in situ SMOD, satellite (SWHLX + Theia) SMOD and residuals (ΔSMOD) stratified by elevation bands in the Pyrenees (left column) and in the Alps (center column). The right column compares both mountain ranges.



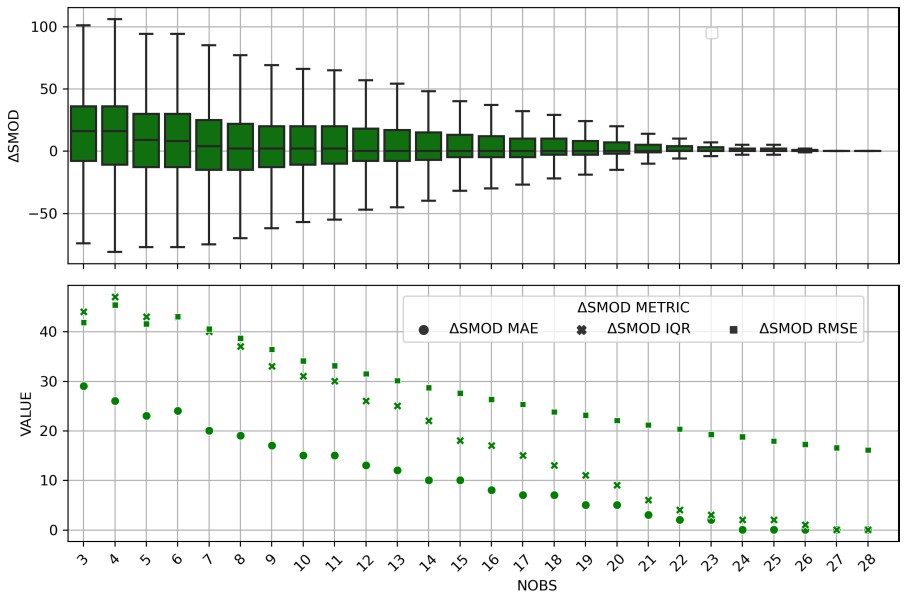

**Figure 8.** Top: ΔSMOD distribution per NOBS from the synthetic dataset derived from Sentinel-2 over the 31TCH tile for the HY 2017. Bottom: ΔSMOD MAE, IQR and RMSE per NOBS from the synthetic dataset derived from Sentinel-2 over the 31TCH tile for the HY 2017.

especially on south and west facing slopes, in massifs such as Haut-Var Haut-Verdon and Aspe Ossau.However, we did not
find a clear influence of aspect or DAH on the distribution of the SMOD trends. There were too few valid SMOD trends in
north and east facing slopes to report a value for these classes.

We found that the trends were sensitive to the period of computation (start and end year), but the impact was more evident in
the Pyrenees. We show here the Mont-Blanc as an example for the French Alps (Fig. 11) and the Aran massif in the Pyrenees
12. In the Mont-Blanc massif, the trends can become positive (non-significant) for periods starting between 1992 and 1999 and
ending between 2011 and 2018. In the Aran massif, at 1500 m and 1800 m, we find positive (significant) trends for periods
ending before 2020. Above 1800 m, and similarly to the Alps, we found an interval of (non-significant) positive trends with
starting years near 1995.





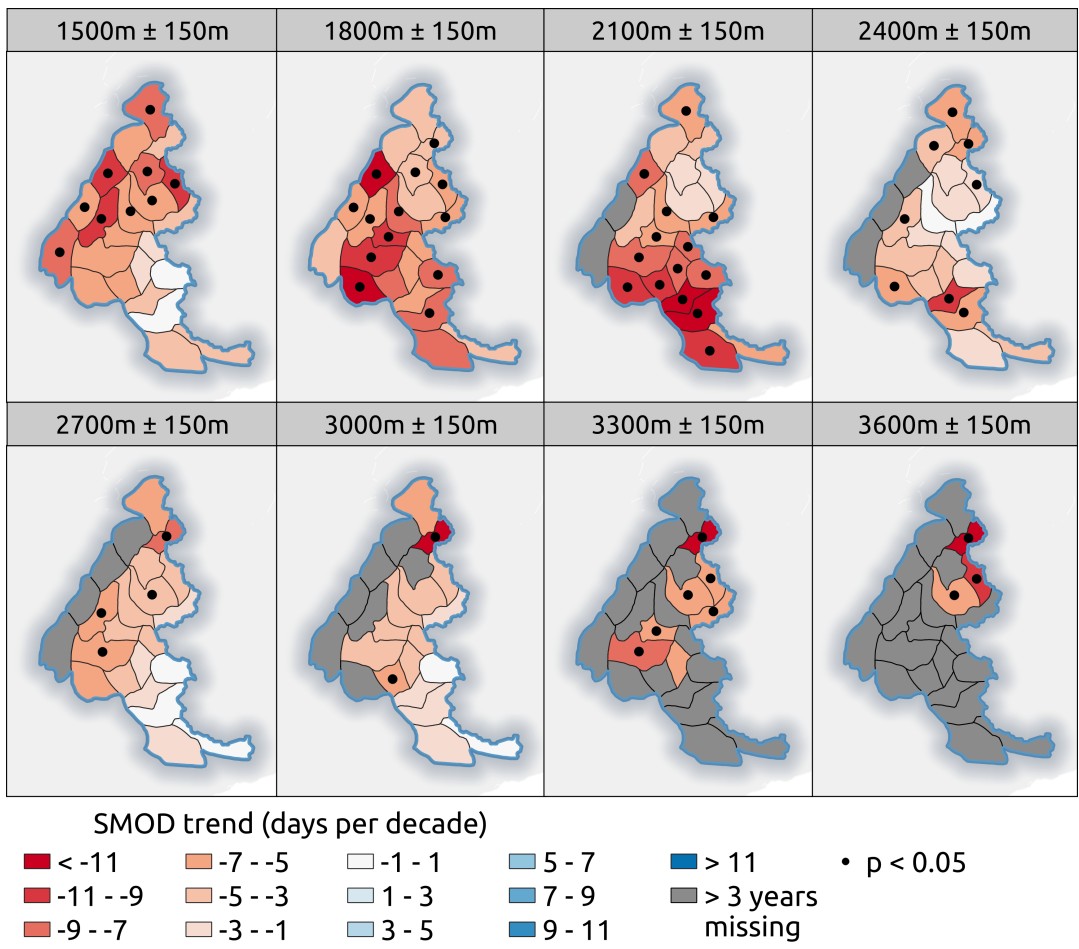

**Figure 9.** French Alps massif-wise SMOD trends for the HY 1986 to 2022. SMOD values were aggregated from $\geq 1000$ pixels with $\text{NOBS}_{\text{min}} \geq 10$. Areas are grayed when less than 35 years of SMOD data were available due to not having enough pixels or to non-existing elevation bands. Dots identify statistically significant trends (p-value < 0.05).



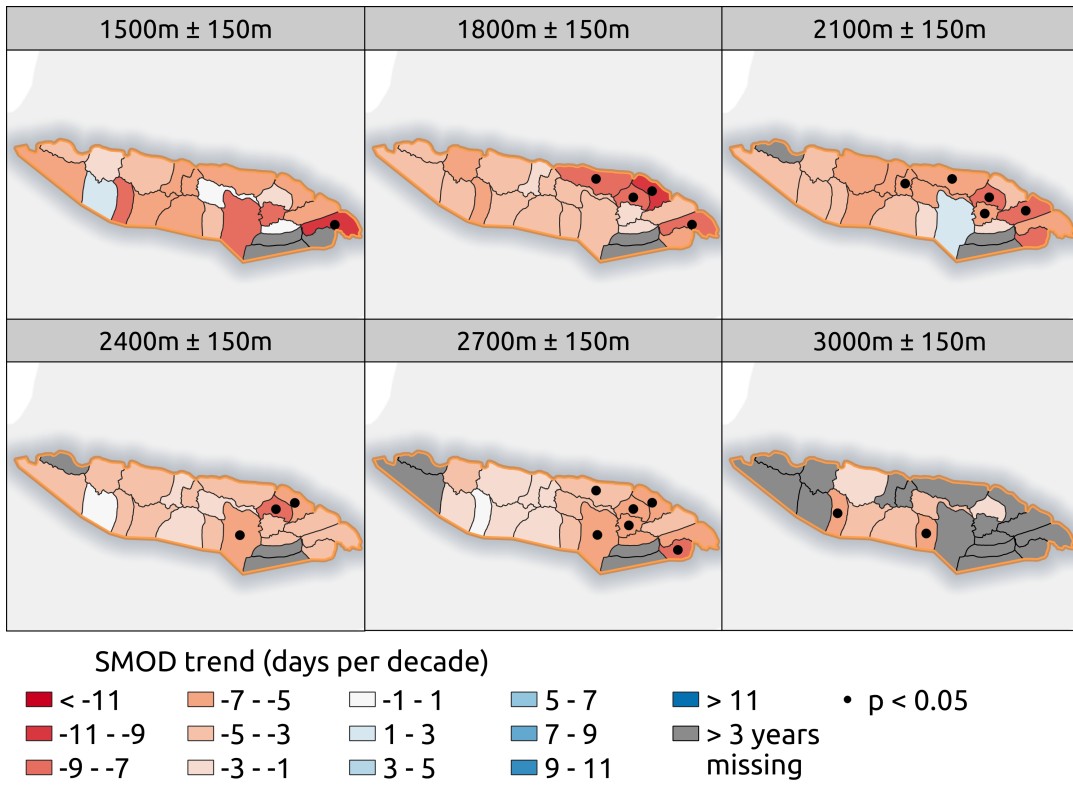

**Figure 10.** Pyrenees massif-wise SMOD trends for the HY 1986 to 2022. SMOD values were aggregated from $\geq 1000$ pixels with $\text{NOBS}_{\text{min}}$ $\geq 10$. Areas are grayed when less than 35 years of SMOD data were available due to not having enough pixels or to non-existing elevation bands. Dots identify statistically significant trends (p-value < 0.05).



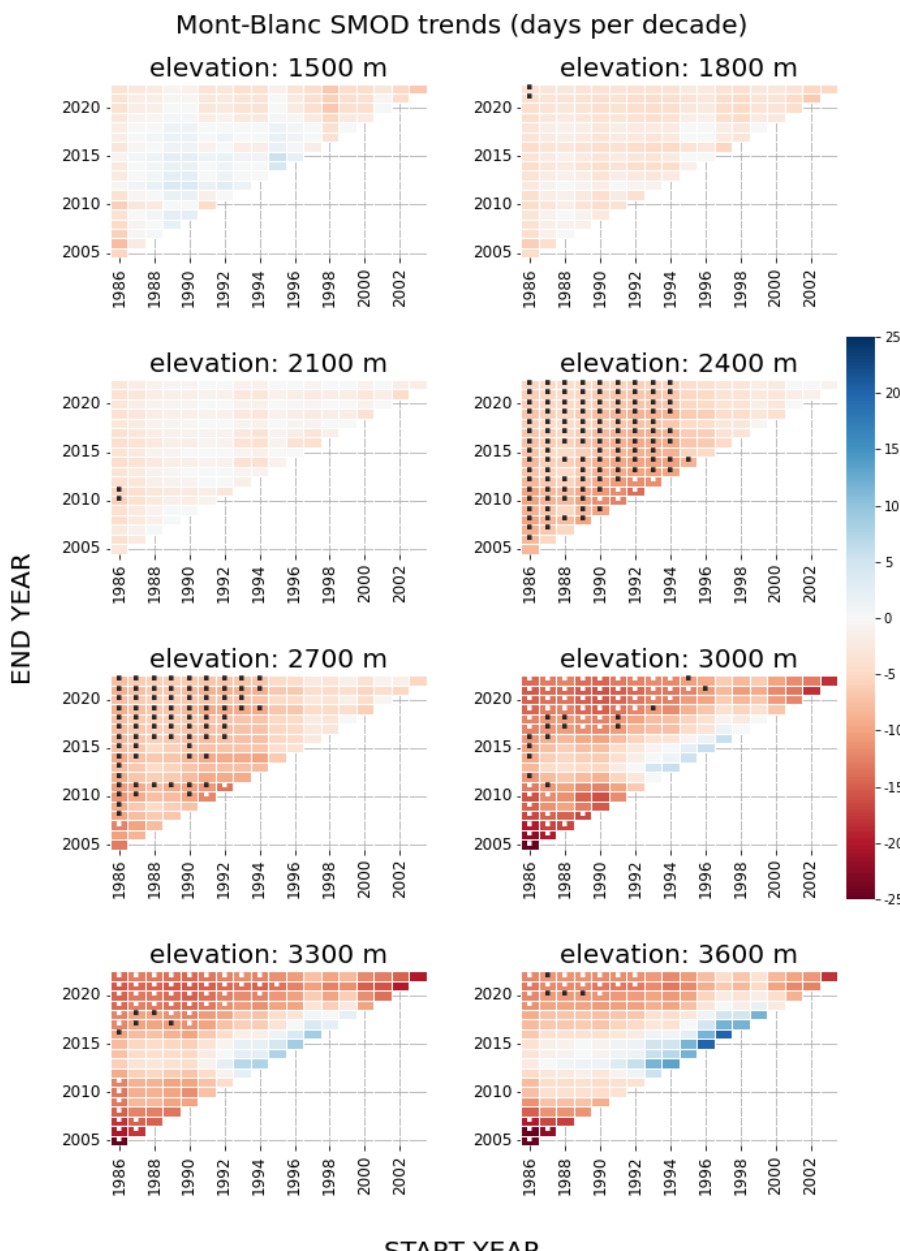

**Figure 11.** Mont-Blanc massif (French Alps). Results of the Mann–Kendall test over the trend of the SMOD median of each elevation band (±150) filtered for $NOBS_{min} \geq 10$ and applied to all possible combinations of start and ending dates involving at least 20 years duration for the HY 1986 to 2022. Colors indicate the SMOD trend as days par decade, and dots indicate statistically significant trends (p-value < 0.05).



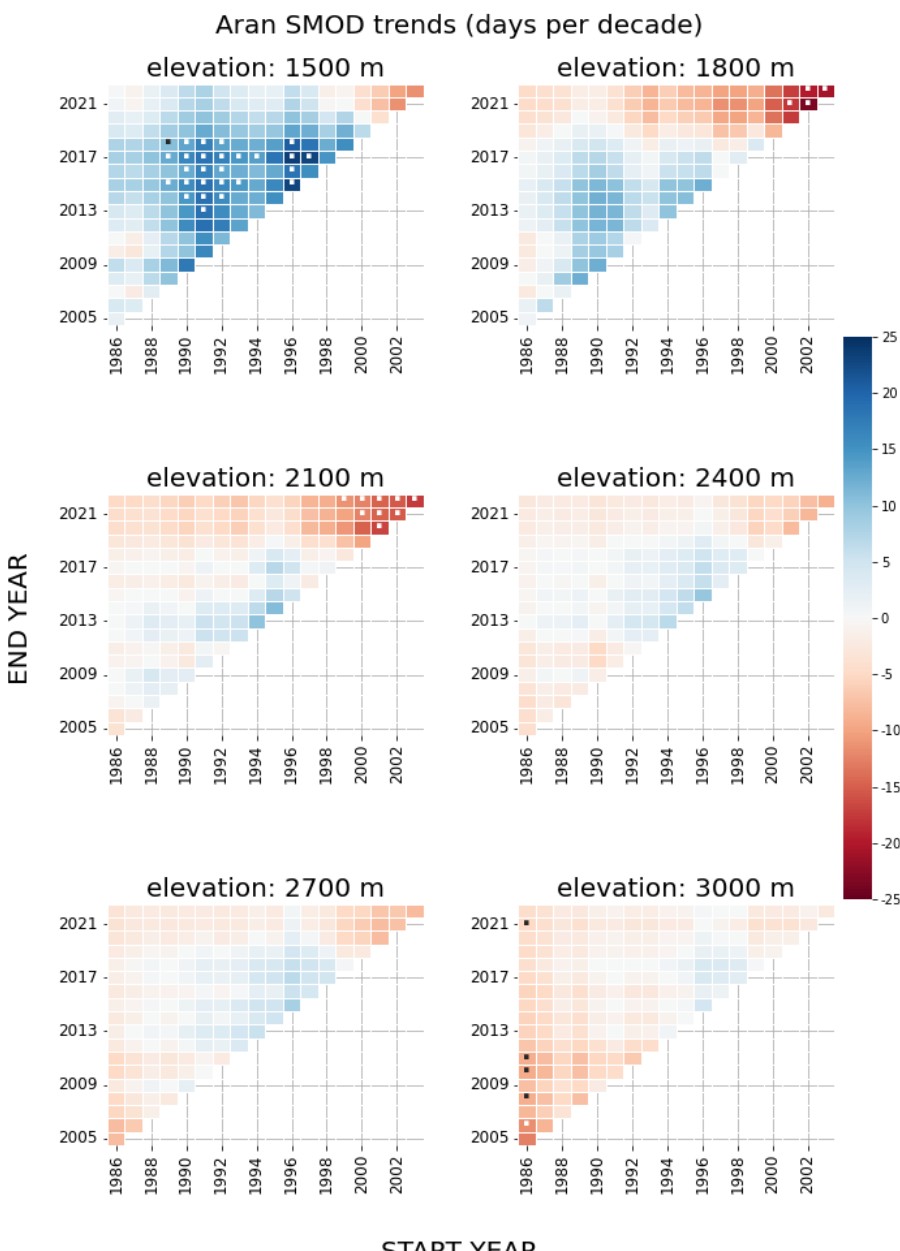

**Figure 12.** Aran massif (Pyrenees). Results of the Mann–Kendall test over the trend of the SMOD median of each elevation band (±150) filtered for NOBS$_{\min} \geq 10$ and applied to all possible combinations of start and ending dates involving at least 20 years duration for the HY 1986 to 2022. Colors indicate the SMOD trend as days par decade, and dots indicate statistically significant trends (p-value < 0.05).





## 5   Discussion

### 5.1   Snow Melt-out Day errors

The satellite SMOD was in relatively good agreement with the in situ SMOD, with half of the ΔSMOD below 11 days. We found an overestimation of the SMOD similarly to previous works using MODIS data. Dietz et al. (2012) observed larger SMOD overestimations than underestimations from daily MODIS time series over Europe and the same was observed over the Tianshan Mountains (Wang and Xie, 2009). Notarnicola (2020) also reported positive biases at low and medium latitudes (including the Alps and Pyrenees) while showing negative biases at high latitudes, similarly to Lindsay et al. (2015) who
found negative biases over Alaska. The main hypotheses mentioned to explain the biases were the uncertainties caused by cloud pixels delaying the observation of snow or no-snow days and the impact of MODIS medium resolution pixels covering heterogenous mountain topography and snow/no-snow transition areas as well as forested and urban areas close to the ground stations used for validation. Positive biases were also found over European mountains using lower resolutions (Metsämäki et al., 2018). Their histogram of ΔSMOD values between 0.1° resolution optical images and 25 km microwave images was
distributed similarly (a mode at zero and positive values reaching up to 100 days) to one shown above with a similar RMSE.

The French Alps showed no negative or positive bias. In contrast, the French Pyrenees showed positive biases at elevation bands from 1500 m to 2100 m and a smaller negative bias at 2400 m. While the in situ data at the 2400 m elevation band were limited and should be treated with caution, both the Alps and the Pyrenees data hint that the SMOD error is reduced at higher elevations. Our hypothesis for the lower performances over the Pyrenees compared to the Alps is that the mountain
range suffers from both a reduced number of high elevation stations and a smaller proportion of NOBS above or equal to 25. The proportions of NOBS $\geq$ 25 are 77.4% for 1201 values over the Alps and 61% for 308 values over the Pyrenees (Fig. 13).

We note that the errors discussed above were obtained at the pixel level hence they represent an upper bound of the error on the SMOD values that were eventually used for the trend analysis, since we performed a spatial aggregation of at least 1000 pixel values before computing trends (section 3.4).

SWH enables to partly overcome the lack of cloud-free observations when using Landsat data only, especially in areas where Landsat revisit times are high (Barrou Dumont et al., 2023). However, it should be noted that this benefit is spatially variable due to the on-demand acquisition mode of SPOT satellites. In the Pyrenees, for example, we observed that SWH acquisitions were more frequent over the French territory.

### 5.2   Snow Melt-out Day trends

A low amount of statistically significant trends, as well as a presence of positive trends at elevation bands of 1500 m and 2100 m for shorter uninterrupted periods, were also observed in the Pyrenees by López-Moreno et al. (2020) between 1986 and 2018. They specifically found an increase of the snow duration at the Aspe-Ossau massif and an increase of the snow depth at Navarra at 1500 m and at Luchonnais at 2100 m. Those observations coincide for Aspe-Ossau and Navarra with satellite-based positive SMOD trends calculated for periods of 20 years or more over the same areas (Fig. C11 in appendix). They do not coincide
with the satellite-based neutral and negative trends calculated at Luchonnais at 2100 m.



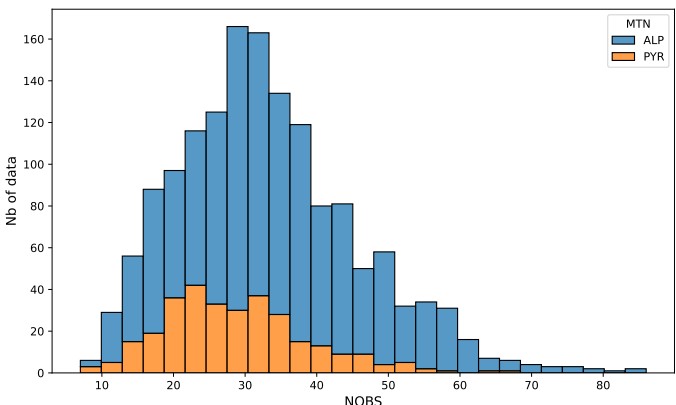

**Figure 13.** Distribution of NOBS values between the stations of the Alps and the stations of the Pyrenees.

Notarnicola (2022) found, for the 1982-2020 period over the Pyrenees, a small (<5%) non-significant increase of the SCA between December and May which is consistent with the positive trends of SMOD seen at low elevations. The overall absence of significant trends over the Pyrenees for the 1986-2023 period can be explained by the influence of the Mediterranean climate over the southern and eastern Pyrenees, which is characterized by a large interannual variability. Despite a well-identified

warming trend (OPCC-CTP, 2018), natural climate variability and in particular winter precipitation variability can mask the effect of warming on the snow cover duration. In particular the variability in snowfall is partly controlled by the North Atlantic Oscillation at decadal timescales (Lemus-Canovas et al., 2024).

We found mostly negative SMOD trends in the French Alps, ranging between -3 and -10 days per decade, in agreement with previous studies in the European Alps over similar time periods using in situ data (Klein et al., 2016; Matiu et al.,

2021), reanalyses (Monteiro and Morin, 2023) or satellite observations (Notarnicola, 2022). We found significant trends at high elevation, unlike previous works using coarser resolution remote sensing data (Hüsler et al., 2014), which highlights the value of high resolution remote sensing in this scope. In particular, negative trends are significant and strongly negative for elevation above 2100 m in the Mont-Blanc massif, which is consistent with previous works (CREA, 2024). Negative trends are stronger in the northern Alps at 1500 m and in the southern Alps at 2100 m. This was also observed by Durand et al. (2009) for

the 1958-2005 period. This spatial gradient is consistent with the Alps climate, as seasonal snow is found at higher elevation in the southern Alps. Our dataset also show that trends in the Mont-Blanc massif were positives (non-significant) for periods starting between 1992 and 1999 and ending between 2011 and 2018 and negatives for periods starting at 2000 and after. This illustrates the importance of using remote sensing time series spanning multiple decades to draw robust conclusions regarding snow cover trends in European mountains. However, our dataset still remains too short to characterize some multidecadal

climatic variations. In particular, in the Alps, Durand et al. (2009) and Marty (2008) hypothesized the presence of a break year in the mid-1980s marking an abrupt transition between two periods with the first having significantly higher snow cover



duration and snow depth than the second. A similar conclusion was drawn from recent reanalyses (Monteiro and Morin, 2023; Beaumet et al., 2021). Our study period either started at or after this break year and thus does not capture this transition.

## 6 Conclusions

The objective of this work was to analyze SMOD trends in the French Alps and Pyrenees and their spatial variability. We combined different collections of SPOT, Landsat and Sentinel-2 products to create an unprecedented time series of high resolution snow cover products. We used this dataset to compute 20 m resolution SMOD images for 37 hydrological years from September 1986 to August 2023. Despite numerous challenges in terms of image processing, temporal gapfilling and data fusion, the estimated SMOD were in agreement with SMOD data obtained from in situ snow depth time series. This allowed

us to estimate SMOD trends from the HY 1986 to 2022 for different regions by topographic class (elevation, aspect, DAH).

We found a general reduction in the SMOD revealing a widespread trend toward earlier disappearance of the snow cover with an average reduction of 5.51 days per decade over the French Alps with a standard deviation of 4.85 days per decade. Over the Pyrenees, we found an average reduction of 4.04 days per decade with a standard deviation of 4.45 days per decade. The results were less homogeneous in the Pyrenees where we also found few positive trends at 1500 m but these trends were not robust

to changing period. Overall, the results were consistent with the previous studies over those mountain ranges. Yet, the studied period might not be long enough to detect trends in areas of short-lived snow cover ("marginal snowpacks" (López-Moreno et al., 2024)) and in areas under the influence of the Mediterranean climate. Other points of interest are:

- In the French Alps, there is a transition of the statistically significant negative trends from the north at 1500 m to the south at 2100 m. This may reflect the climatic gradient like the increasing elevation of the zero-degree isotherm.

- There is a strong correlation between the pixel-wise SMOD uncertainties and the NOBS when the latter is under 26, and a linear function of NOBS could be used to model the SMOD error and generate a pixel-wise quality mask accompanying the SMOD products for more local applications.

- 20 m resolution pixels give enough data points to build SMOD medians robust to the uncertainties from lower NOBS, and to build significant trends from deeper levels of stratification (massif, elevation, aspect/DAH).

Other snow phenology variables like the snow onset day or the snow cover duration could also be estimated using the same dataset. However, the snow onset day is probably subject to higher uncertainty due to the presence of clouds when the snow season begins. Ultimately, the snow cover products (level 2B) could be used to evaluate and/or constrain climatological reanalyses of the snow cover through data assimilation in combination with in situ data and existing meteorological models. For that purpose, the error observation could be estimated from our results as a function of the number of available observations.

Finally, the same satellite data could be used to study vegetation trends using the near-infrared band of the images.



*Code availability.* The level 3B processor which computes the SMOD from satellite image timeseries is available from the let-it-snow repository https://gitlab.orfeo-toolbox.org/remote_modules/let-it-snow/

*Data availability.* The SMOD images produced for this study are available from (Barrou Dumont et al., 2024a). The SMOD time series are updated every year using Sentinel-2 products and distributed as level 3B products of the Theia Snow collection (Gascoin et al., 2019).

*Author contributions.* ZBD and SG wrote the manuscript. ZBD and SG processed data and analyzed the results. SG and JI supervised the research. JK and AD processed Landsat-DLR products. DM and CC prepared in situ data. All coauthors gave feedback on the results and edited the manuscript. All coauthors have read and agreed to the published version of the manuscript.

*Competing interests.* We declare that no competing interests are present.

*Acknowledgements.* This work was supported by the ANR TOP project, grant ANR-20-CE32-0002 of the French Agence Nationale de la
Recherche. We aknowledge support from CNES for the processing of SPOT 1C data the development of the software used for the SMOD estimation. We thank Samuel Morin for his contribution to the in situ data processing and his thoughtful review of our manuscript.





**Appendix A: Snow and cloud classification in historical SPOT images**

Barrou Dumont et al. (2024b) developed an emulator of SPOT images from Sentinel-2 1C images to train a statistical model of snow and cloud classification in SPOT images. Indeed, the emulation approach allowed pairing a pseudo-SPOT image with a Sentinel-2 level 2B product which provides the labels (snow, no-snow, cloud) required for the training. The trained model was a U-net convolutional neural network (Ronneberger et al., 2015), which yielded high precision in detecting snow and minimal false snow pixel identification. However, haze and high transparent and semi-transparent clouds could be detected in Sentinel-2 images thanks to their higher number of spectral band and their radiometric quality. These clouds were almost invisible in pseudo-SPOT images, creating "false" clouds in the training dataset and causing an overestimation of cloud pixels by the U-net.

To solve this issue, we filtered the cloud pixels from the training data according to how they were detected. The cloud mask in level 2B products was generated with the MAJA software which provides information on the way that a cloud pixel was detected (Hagolle et al., 2017). In particular, one way to detect cloud was a combination of pixel-wise mono-temporal reflectance thresholds in the blue, red, NIR, and SWIR bands. Cloud pixels detected with the mono-temporal threshold were certain to be also visible in a SPOT 1-5 instrument. Keeping only those cloud pixels reduced the amount of misleading labels.

Cloud shadow pixels were also removed from the training to capitalize on the observation from Barrou Dumont et al. (2024b) that the U-net was able to detect snow in less illuminated areas.

We also changed the number of times the training dataset was run through the neural network (number of epochs). In Barrou Dumont et al. (2024b), the training starts with multiple short parallel preliminary trainings of 40 epochs to look for the best weights initialization where the U-net can converge to a state of minimum loss, and continues with a more intensive training of 200 epochs. A solution to ensure that the U-net converged to the state of minimum loss afforded by its architecture and the training data was to remove the limit over the number of epochs and stop each training when the U-net stops improving for a given amount of epochs (40 for a preliminary training, 200 for an intensive training).

One U-net model was trained for each of five SPOT instruments and by mountain range (French Alps, Pyrenees), i.e. a total of 10 models. The SWIR band in SPOT 4 HRVIR and SPOT 5 HRG was emulated and included as an additional input in the training of the SPOT 4 and SPOT 5 models. For the Pyrenees, Sentinel-2 training images were selected randomly with the same methodology of (Barrou Dumont et al., 2024b). For the French Alps, the Sentinel-2 tiles are 31TGM, 31TGL, 31TGK, 31TGJ, 32TLS, 32TLR, 32TLQ, and 32TLP, and only complete images (no no-data in the image edges) were selected (complete images correspond to a relative orbit number of 108 for the French Alps). This represented a total of 46 images over the Pyrenees and 92 images over the French Alps. Inference was then applied with each model to extract snow and cloud cover maps from the entire SPOT dataset. Because the inference of a pixel depends on the value of the surrounding pixels, we masked pixels too close (2000 m) to the border of the image or too close to no-data areas of the image to ensure an equal performance of the neural network across the data.





**Table B1.** Confusion matrix between prediction and observation of snow and no-snow classes.

| $2 \times 2$ confusion matrix | | |
| --- | --- | --- |
| Prediction\Observation | snow | no-snow |
| snow | TP | FP |
| no-snow | FN | TN |

## Appendix B: Statistical metrics

435 The confusion matrix aggregates the different combinations of predicted and observed (for example: ground truths) values. When the prediction is a binary choice (snow or snow-free), the correct prediction of the value of interest (snow) is called True Positive (TP) and it's incorrect prediction is called False Positive (FP). The correct prediction of the alternative value (snow-free) is called True Negative (TN), and it's incorrect prediction is called False Negative (FN) (Tab. B1). Those metrics can then be used to calculate the precision, which is the fraction of correct prediction of snow, the recall, which is the fraction

440 of observed snow which has been retrieved, and the F1 score, which is the harmonic mean of the precision and the recall (Eq. B1). The F1 score is particularly well adapted for evaluating a classifier over an unbalanced dataset, which is to be expected when using satellite images of mountain ranges.

$$
\begin{aligned}
\text{Precision}_{\text{snow}} &= \frac{\text{TP}}{\text{TP} + \text{FP}} \\
\text{Recall}_{\text{snow}} &= \frac{\text{TP}}{\text{TP} + \text{FN}} \\
\text{F1}_{\text{snow}} &= \frac{2 \times \text{Precision}_{\text{snow}} \times \text{Recall}_{\text{snow}}}{\text{Precision}_{\text{snow}} + \text{Recall}_{\text{snow}}}
\end{aligned}
\tag{B1}
$$

Common metrics for the evaluation of continual outputs like the SMOD includes the RMSE which is the quadratic mean of the difference between predicted and observed values. Because the errors are squared, the RMSE is sensitive to outliers, as even a few number of large errors can raise it's value significantly. The Median Absolute Error (MAE) is the median of the differences where the sign of the difference is ignored to avoid cancellations between positive and negative errors. Contrary

450 to the RMSE, the MAE is not affected by outliers. The Interquartile range (IQR) is the difference between the 75th and 25th percentiles of a dataset. The RMSE is expressed as:

$$
\text{RMSE}(y, \hat{y}) = \sqrt{\frac{\sum_{i=0}^{N-1} (y_i - \hat{y}_i)^2}{N}}
\tag{B2}
$$

With $i$ the value number, $N$ the number of values, $y$ the predicted value and $\hat{y}$ the observed value.

## Appendix C: Supplementary figures



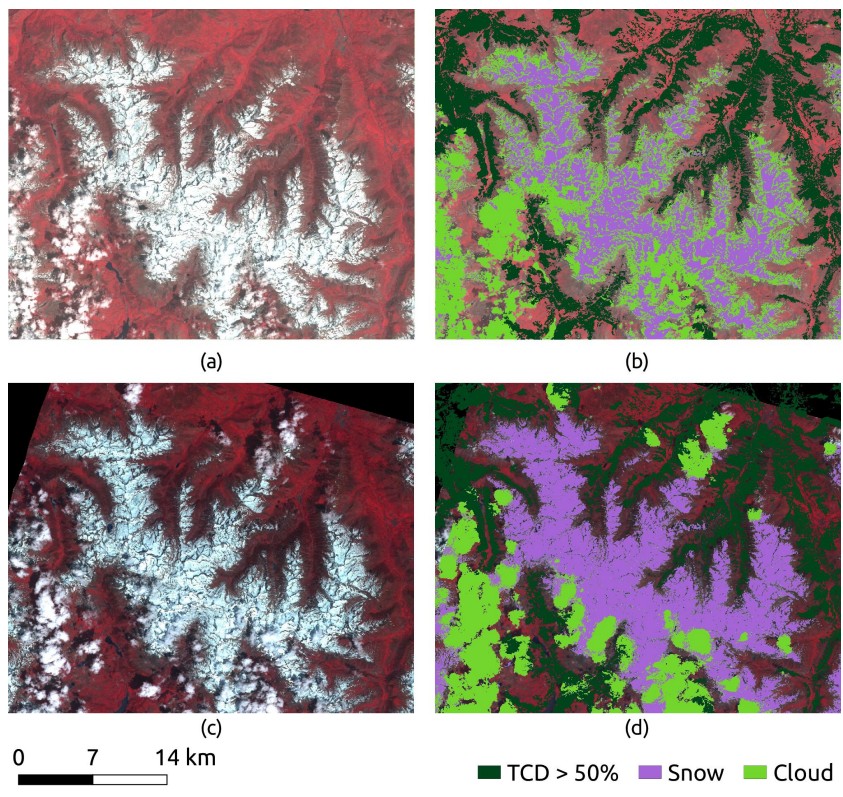

**Figure C1.** Pyrenees 25/05/1987: 30x30 km$^2$ scene with a) Landsat 5 TM image (Green, Red, SWIR); b) DLR-Landsat 2B product; c) SPOT 1 HRV image (Green, Red, NIR); d) SWH 2B product.





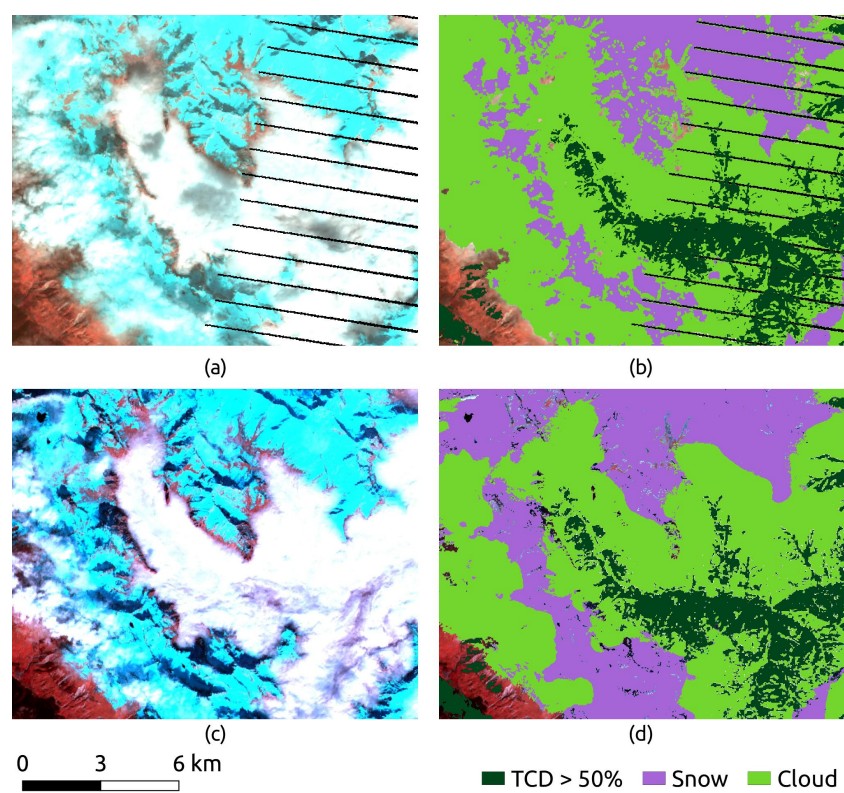

**Figure C2.** Alps 13/10/2005: 12x12 km$^2$ scene with a) Landsat 7 ETM+ image (Green, Red, SWIR); b) DLR-Landsat 2B product; c) SPOT 4 HRVIR image (Green, Red, SWIR); d) SWH 2B product.



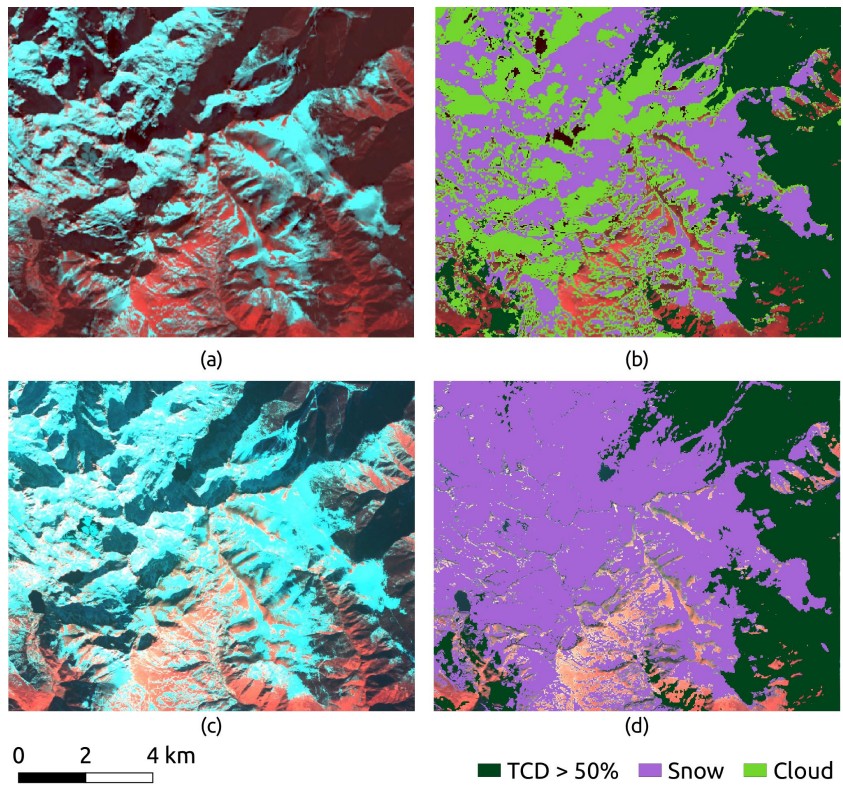

**Figure C3.** Alps 03/12/2013: 12x12 km$^2$ scene with a) Landsat 8 OLI image (Green, Red, SWIR); b) DLR-Landsat 2B product; c) SPOT 5 HRG image (Green, Red, SWIR); d) SWH 2B product.




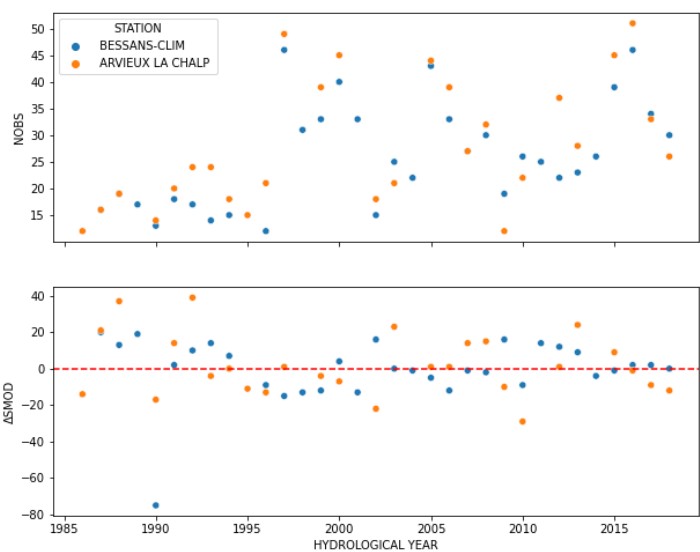

**Figure C4.** Yearly NOBS (top) and SMOD (bottom) values from the two stations which recorded only observed snow melt dates (not gap-filled) for the entirety of the study period.

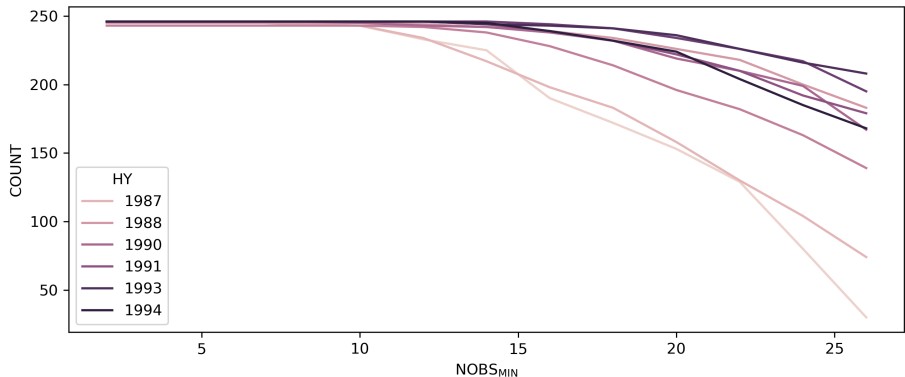

**Figure C5.** For every HYs below 1995, number of available SMOD medians from every combinations of massif and elevation bands as a function of $NOBS_{min}$. A loss of data was observed for $NOBS_{min}$ thresholds above 10, with a stronger effect before 1989. For later periods, a $NOBS_{min}$ of 26 still allows to maintain most of the available combinations of massif and elevation. For 1986, a $NOBS_{min}$ of 26 reduces the available combinations of massif and elevation bands to 28.



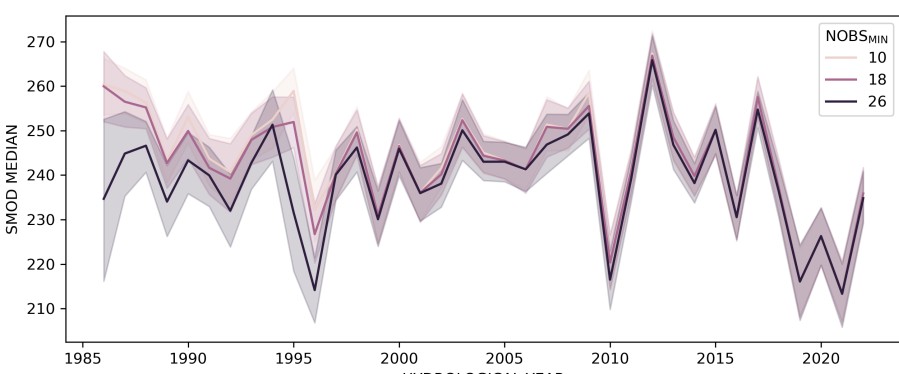

**Figure C6.** For NOBS$_{min}$ of 10, 18, and 26, yearly average of the SMOD medians from each combination of massif and elevation band, with a confidence interval of 95%. The distribution is robust to NOBS$_{min}$ except for earlier years at a NOBS$_{min}$ of 26 due to the significant reduction of available combinations.

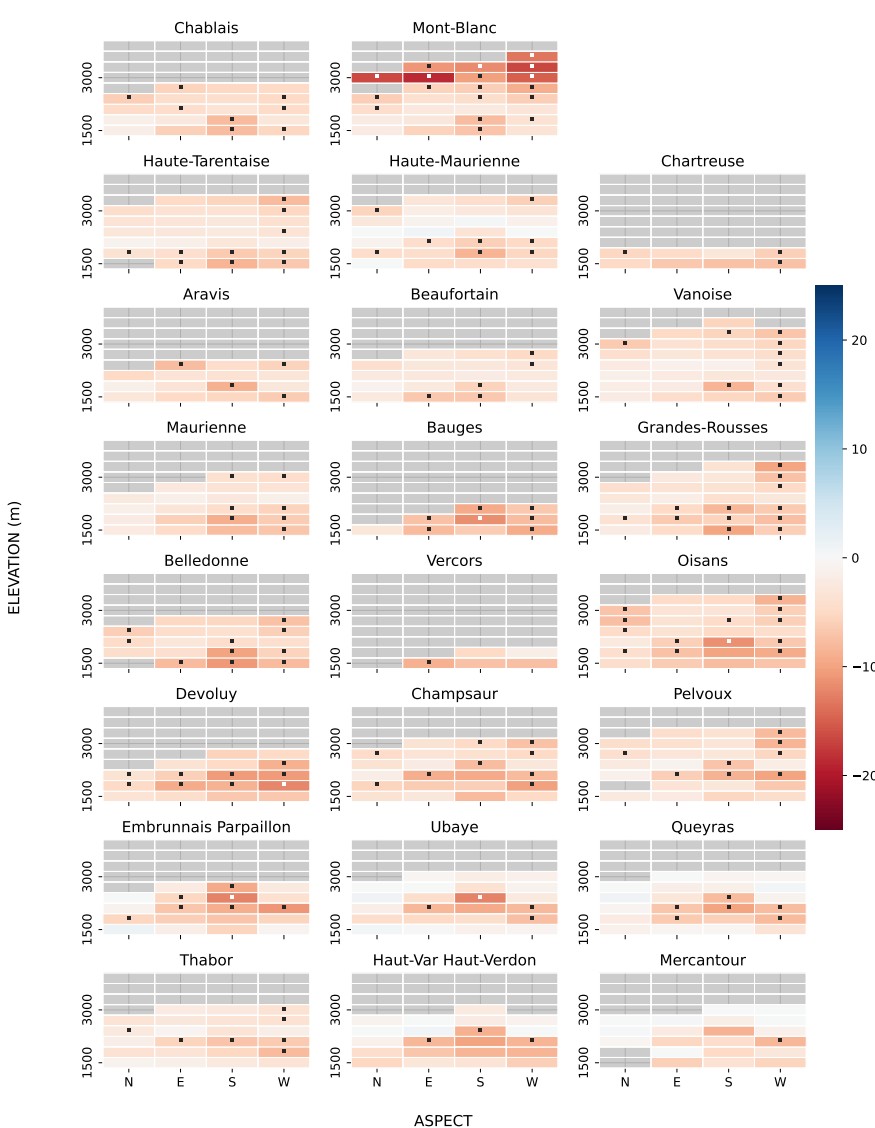

**Figure C7.** French Alps. Heatmaps of the Mann–Kendall test over the trend of the SMOD median of each elevation band (±150) and aspect filtered for NOBS$_{min}$ ≥ 10. Colors indicate the SMOD trend as days par decade, and dots indicate statistically significant trends (p-value < 0.05).

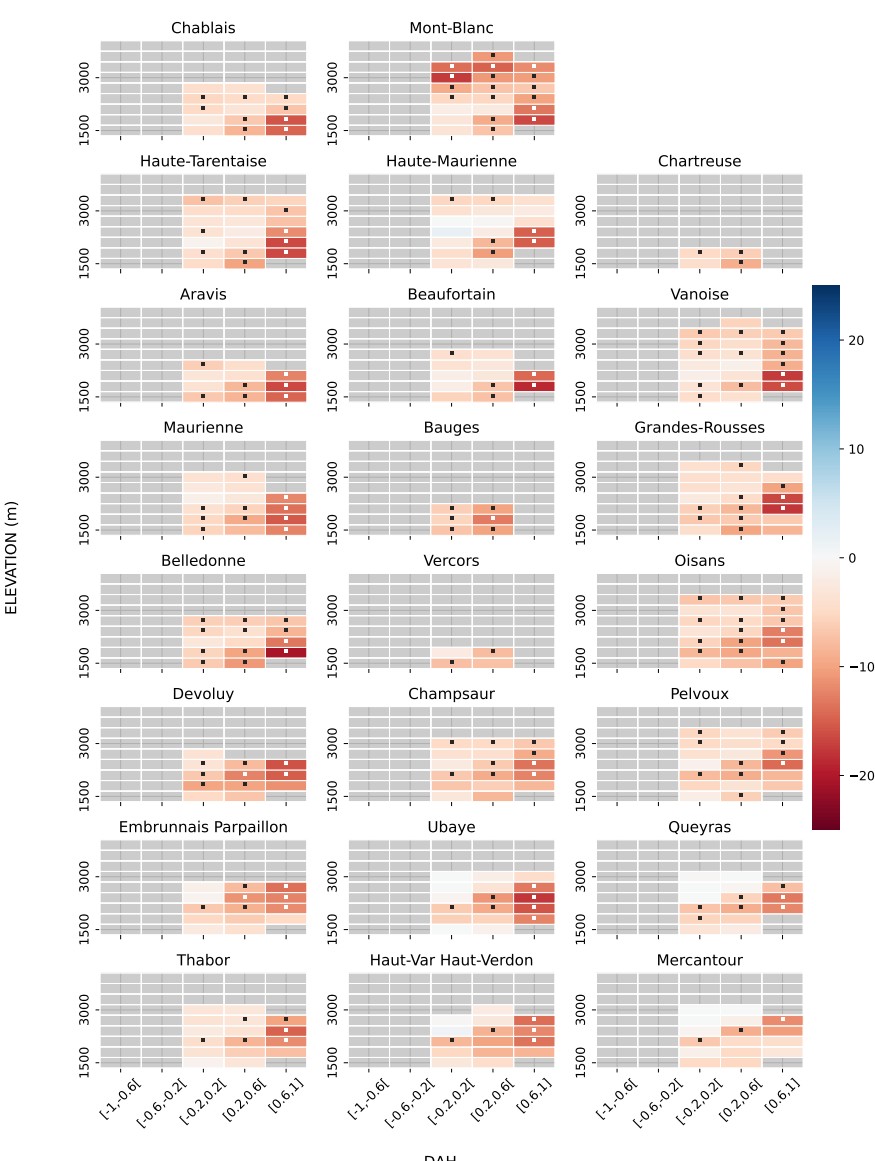

**Figure C8.** French Alps. Heatmaps of the Mann–Kendall test over the trend of the SMOD median of each elevation band (±150) and DAH filtered for $NOBS_{min} \geq 10$. Colors indicate the SMOD trend as days par decade, and dots indicate statistically significant trends (p-value < 0.05).



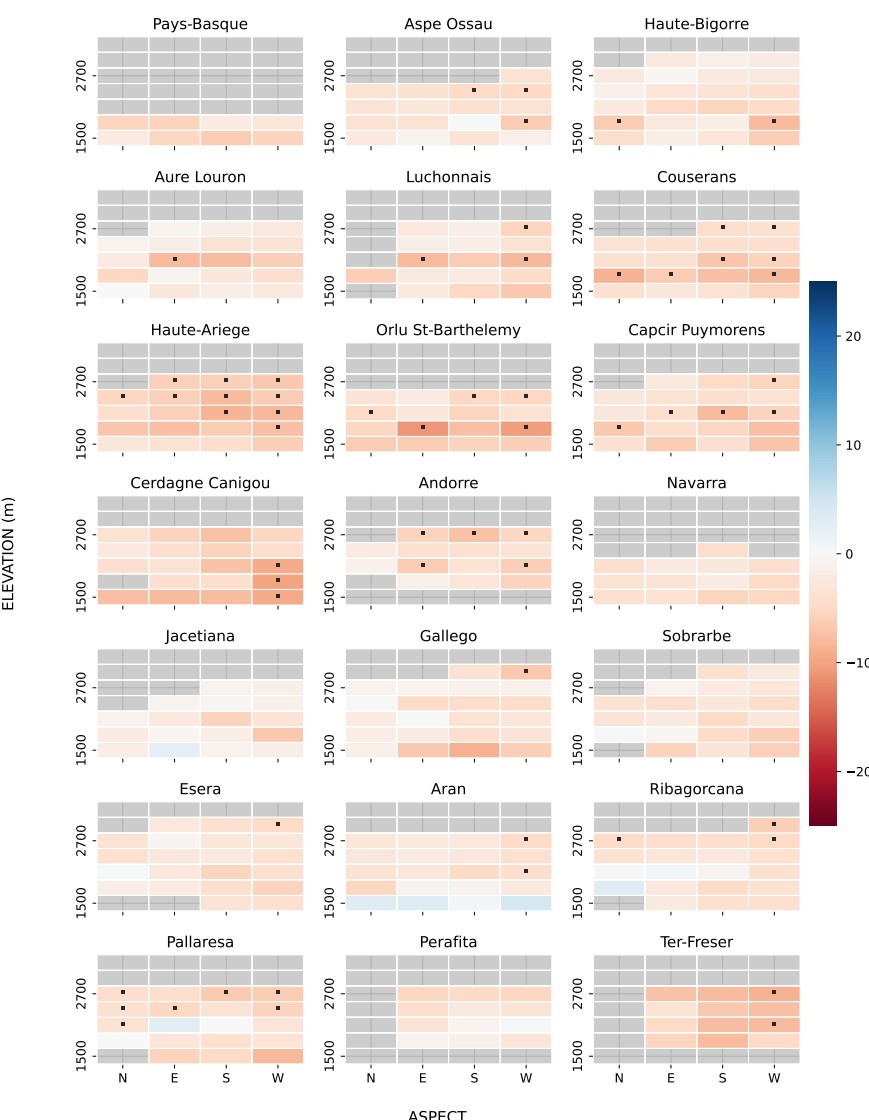

**Figure C9.** Pyrenees. Heatmaps of the Mann–Kendall test over the trend of the SMOD median of each elevation band (±150) and aspect filtered for NOBS$_{min}$ ≥ 10. Colors indicate the SMOD trend as days par decade, and dots indicate statistically significant trends (p-value < 0.05).



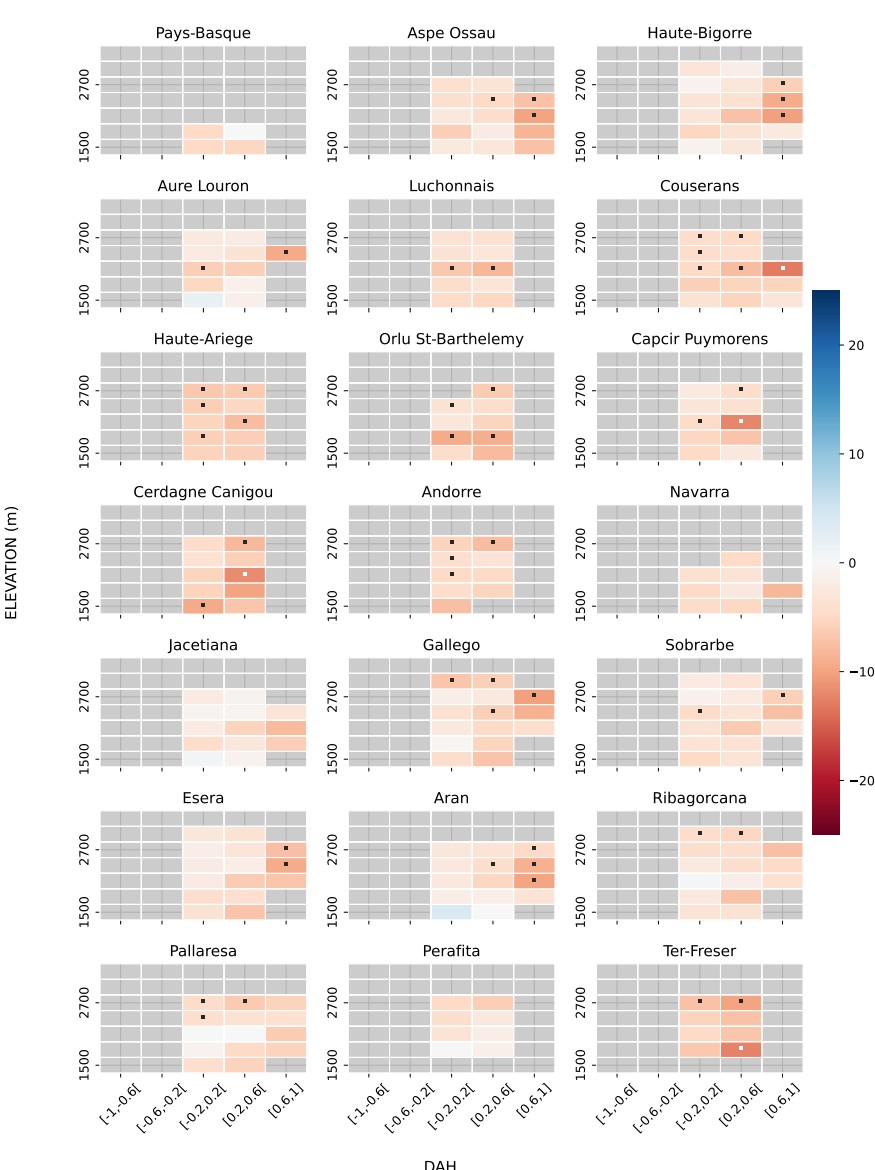

**Figure C10.** Pyrenees. Heatmaps of the Mann–Kendall test over the trend of the SMOD median of each elevation band (±150) and DAH filtered for $NOBS_{min} \geq 10$. Colors indicate the SMOD trend as days par decade, and dots indicate statistically significant trends (p-value < 0.05).





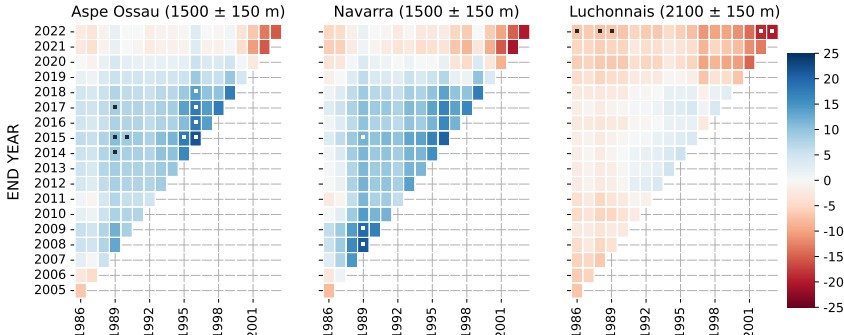

**Figure C11.** Massifs Aspe Ossau (1500 m), Navarra (1500 m), and Luchonnais (2100 m). Results of the Mann–Kendall test over the trend of the SMOD median filtered for $NOBS_{min} \geq 10$ and applied to all possible combinations of start and ending dates involving at least 20 years duration for the HY 1986 to 2022. Colors indicate the SMOD trend as days par decade, and dots indicate statistically significant trends (p-value < 0.05).





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
