# Peer review of "Trends in the annual snow melt-out day over the French Alps and the Pyrenees from 38 years of high resolution satellite data (1986-2023)"

_EGUsphere, 2024_

## Author Comment (AC2)

**Response to Anonymous Referee #2**

[egusphere-2024-3505] Trends in the annual snow melt-out day over the French Alps and the Pyrenees from 38 years of high resolution satellite data (1986–2023)

We thank the referee for the helpful and relevant comments. We explain below how we aim to take them into account in a revised manuscript.

This study uses multi-source datasets to reveal snow melting day over the long time series. Employing various methods, it improves the spatial-temporal resolution of snow melting day to support the study results. Authors have made a comparison between SWH and landsat, but I am also worried about the data consistency between different landsat sensors, and between landsat and sentinel-2, especially under the case with a long study period and the mixture of all available data. So my suggestion is to add some data validation before using satellite datasets to calculate snow melting day.

We agree that data consistency is important. First of all we note that all sensors are multispectral imagers at 20-30m resolution, i.e. we do not blend high and low resolution imagery (like MODIS, AVHRR), or radar (like ERS or Sentinel-1). Secondly, we have evaluated the annual SMOD products that results from our multi-sensor time series and assessed their uncertainty in the paper. In the end, this is what matters for us since our objective is the recent evolution of the SMOD. We did not detect any trend in the mean SMOD error, but a larger spread in the beginning of the study period, that we attribute mainly to the lower revisit. In this paper we focused on the comparison between Landsat-DLR and SPOT SWH products because these products result from different retrieval algorithms (NDSI-based unsupervised classification vs. deep learning). Landsat products before 2018 (Landsat-DLR) were already evaluated by Hu et al. (2019a, 2019b)[1]. The performances of SPOT4 products were also previously evaluated by Barrou Dumont et al. (2024)[2], and the spectral characteristics of the SPOT products are essentially near-identical outside of the addition of the SWIR band to SPOT4 and 5. However, in response to the reviewer we have performed an additional analysis to evaluate the agreement between Landsat and Sentinel-2 products after 2016 as suggested (see below).
* * *
[1] Hu, Z., Dietz, A., & Kuenzer, C. (2019a). The potential of retrieving snow line dynamics from Landsat during the end of the ablation seasons between 1982 and 2017 in European mountains. International Journal of Applied Earth Observation and Geoinformation, 78, 138-148.
Hu, Z., Dietz, A. J., & Kuenzer, C. (2019b). Deriving regional snow line dynamics during the ablation seasons 1984–2018 in European Mountains. *Remote Sensing*, *11*(8), 933.

[2] Barrou Dumont, Z, S. Gascoin and J. Inglada, "Snow and Cloud Classification in Historical SPOT Images: An Image Emulation Approach for Training a Deep Learning Model Without Reference Data," in *IEEE Journal of Selected Topics in Applied Earth Observations and Remote Sensing*, vol. 17, pp. 5541-5552, 2024, https://doi.org/10.1109/JSTARS.2024.3361838

Line 110, when you combined the landsat data from different sensors over the long time series, how did you consider the sensor correction or data consistency between products from various sensors? In addition, can you give a brief description regarding the algorithm to produce level 2B product?

We retrieved the snow cover area from Level-2 Landsat Collection 2 Level-2 that are processed by USGS to provide consistent surface reflectance across different Landsat sensors and thus enable time series analysis (Köhler et al. 2022[3]). The USGS applies radiometric calibration and atmospheric correction algorithms to Level-1 Landsat data. We can give a brief description of the Level-2B algorithm (snow detection) in a revised manuscript.

Line 115, before mixing sentinel-2 and landsat-8 data from 2016 to 2023, I miss the consistency check between both datasets or sensor correction because they are from different sensors. Only if the data from different sensors are proven to be consistent, the trends calculated later will be meaningful.

We performed a comparison between eight pairs of snow products from Landsat 8 (L8) and Sentinel-2 (S2). These image pairs were acquired on the same day between November 2017 and June 2018, covering areas in the Alps and the Pyrenees.

Products were filtered using the TCD mask, the water mask, and the glacier mask, to exclude pixels that are not used in the SMOD trend calculation. Additionally, pixels containing clouds detected in either S2 and/or L8 were excluded from the analysis since images were not acquired at the same time of the day.

The analysis, conducted on a dataset of $77 \times 10^6$ pixels, yielded the following results:

34% of the pixels were classified as snow by S2.
31% of the pixels were classified as snow by L8.
S2 detected 96% of the snow pixels identified by L8.
L8 detected 89% of the snow pixels identified by S2.
The F1 score between the two products was 0.92.

We will include these results in a Supplement of the revised manuscript.

Line 183, what is the basis to set the threshold of tree cover 50%?

We aimed to exclude dense forests where the snow detection is very uncertain with multispectral optical sensors such as Landsat and Sentinel-2 - without excluding sparse forests where the snow detection is still accurate (Muhuri et al. 2021[4], Barrou Dumont et al.

[3] Koehler, J.; Bauer, A.; Dietz, A.J.; Kuenzer, C. Towards Forecasting Future Snow Cover Dynamics in the European Alps—The Potential of Long Optical Remote-Sensing Time Series. Remote Sens. 2022, 14, 4461. https://doi.org/10.3390/rs14184461

[4] Muhuri A., Gascoin, S., Menzel, L., Kostadinov, T., Harpold, A., Sanmiguel-Vallelado, A., López-Moreno, J. I., (2021) Performance Assessment of Optical Satellite Based Operational Snow

2021[5], Figure 3). We note that the TCD distribution across our study domain is bimodal with a mode near 0% and a mode near 100%. Therefore we expect the sensitivity to this value to be low.

Line 199, what is NOBS?

This variable refers to the number of observation and was indeed not defined in the main text, we apologize for this omission that we will correct in the revised article.

Check the figure 5. The left sub-figure is landsat only or landsat+spot?

There was indeed a mistake in the caption. The left panel is SPOT+Landsat.

Fig. 9 and Fig. 10, how about mapping pixel-wise? One of advantages of this study is high spatial resolution.

We took advantage of the spatial resolution by stratifying the analysis in 300 m elevation bands up to 3600 m high. The high resolution allowed us to obtain a large number of pixels even at high elevations, which would not be possible with coarser resolution sensors. In addition, the aggregation by elevation and massif contributes to reduce the noise at the pixel level (random error) and thus to show more robust results. A map at 20 m resolution would have to be resampled to a much coarser resolution to fit in the article format (without resampling at 72 dpi, a 300 km wide domain such as the Pyrenees measures approximately 5 m wide). This is why we choose to show only a small region at full resolution (Figure 5). Finally, the products are available at full resolution in a public repository.

Line 322, I am not sure how tree cover density correlates to elevation? Maybe the changes in bias with elevation could be partly explained by tree cover density. How about dividing tree cover density into several classes? Then analyze something, such as the bias and trend.

Following this suggestion we have computed the correlation of TCD and elevation from all the pixels corresponding to elevations > 1200m and TCD > 0 and TCD < 50. We obtained a correlation of -0.01 between TCD and elevation (therefore no correlation between the two). We will add this point to the Discussion.
* * *
Cover Monitoring Algorithms in Forested Landscapes, IEEE Journal of Selected Topics in Applied Earth Observations and Remote Sensing, https://doi.org/10.1109/JSTARS.2021.3089655.

[5] Barrou Dumont, Z., Gascoin, S., Hagolle, O., Ablain, M., Jugier, R., Salgues, G., Marti, F., Dupuis, A., Dumont, M., and Morin, S. (2021) Brief communication: Evaluation of the snow cover detection in the Copernicus High Resolution Snow & Ice Monitoring Service, The Cryosphere, 15, 4975–4980, https://doi.org/10.5194/tc-15-4975-2021

---

## Author Response (AR1)

**Response and changes**

[egusphere-2024-3505] Trends in the annual snow melt-out day over the French Alps and the Pyrenees from 38 years of high resolution satellite data (1986–2023)

**Anonymous Referee #1**

**Comment 1:**

**Referee:** *Introduction.* The objective of the paper does not appear clearly. Please add more information about it in the introduction. The link between second and third paragraph (L32-L33) could be improved. For example, presenting first the different options that can be use for SMOD trends (numerical modeling, in situ observations and satellite data), as it is done at L46-56, and once the advantages of satellite data highlighted, describe the available products (L56-84). This can now lead to your interest in merging different sources of data to create the most suitable dataset for studing SMOD trends. Moreover, last paragraph already presents some methodology and the studied domains with details, which, in my point of view, should be in the Method section. The introduction could also end with an overview of the structure of the paper.

**Authors' response:** We understand this presentation is a bit disturbing but we would like to keep it because we feel that scientific objectives should motivate the technical choices. We identified a knowledge gap regarding the trends of SMOD in European mountains. We state the scientific requirements to produce a meaningful analysis (30 years minimum, 100 m resolution). Then we review the available datasets (in situ, model, satellite). Given the different constraints of each data sources, satellite data appears as the logical choice. We will reformulate this section to improve the logical flow for the reader. The last paragraph can indeed be modified to remove some methodological details but we think that at least the study area (in km2) should be kept here as it shows that our study covers a much larger domain than previous studies. We will move Fig. 1 to the next section as suggested below.

**Changes**: We kept the structure of the introduction but made the objective clearer. The introduction to massifs and Fig1 were moved in method (section 3.3)

**Comment 2:**

**Referee:** *Data and methods.* I think that some reorganisation would be beneficial for a better understanding. In the result part, two sets of data are considered: SWHLX and Theia. My suggestion would be to divide section 2.1 in two parts: SWHLX (2.1.1) and Theia (2.1.2). In 2.1.1, you could present SWH and DLR-Landsat, the main conclusions of your evaluation, and finally the SWHLX dataset. Depending our main objective (SMOD trends or building a dataset and use it for SMOD trends), you could put in Annexe the method and results of the evaluation (in the present version: 3.1 and 4.1) or let it in the 2.1.1.

**Authors' response:** We find that this is a good suggestion to better organize the presentation of the data and we agree to follow it in a revised manuscript (i.e. merge 2.1.1 with 2.1.2 and create two subsections SWH and DLR-Landsat). However we cannot present the results of our evaluation in this Data section because we need to present some methodological details. The best option is to move the full package (method and result) of "Evaluation of SWH and DLR-Landsat agreement" to a supplement document. Then the article would really focus on the SMOD trends as suggested and that would also help address the main general comment.

**Changes:** The data section was changed as asked and "Evaluation of SWH and DLR-Landsat agreement" was moved in the appendix.

**Comment 3:**

**Referee:** Why did you select this specific tile (31TCH) and HY (2017)? Is it a method usually applied in such study?

**Authors' response:** We focused on a single tile because the method is computationally intensive. We chose this tile in the Pyrenees because it contains a large topographic and land cover variability. We used it for benchmarking in previous studies and have a good knowledge of the Theia products in this area (see e.g. Gascoin et al. 2020). We chose HY 2017 because both Sentinel-2 and Landsat 8 data were available from Theia (Landsat products are no longer distributed after 2020). This method is new to our best knowledge. We designed it specifically for the purpose our this study. We used a similar idea (using Sentinel-2 to emulate older satellite archives) in a previous study (Barrou Dumont et al. 2024)

**Changes:** no changes

**Comment 4:**

**Referee:** In Fig. 4, no example are shown for an odd number of days without data between a no-snow and a snow day. Can you elaborate on this point, please?

**Authors' response:** We thank the referee for this remark. If there is an odd number of days without data between a no-snow and a snow day, the central day is a tie. The code returns the snow value. We chose this tie-breaking rule (rounding toward the snow class) because the snow detection algorithm tends to assign a no-snow class to the lowest snow cover fraction. We will clarify this in the manuscript.

**Changes:** Figure was changed to show odd number cases and a clarification was added in the method section "Calculation of the snow melt-out day"

**Comment 5:**

**Referee:** Two stations were selected only for their data availability, and no more information were provide about their characteristics. Is there any characteristics of these stations that can explain the absence of a specific biais between in situ and satellite data?

**Authors' response:** Both stations are situated in the french alps, where an absence of SMOD biases has also been observed compared to the Pyrenees. They also both generally have NOBS > 10 throughout the study period, which could appear from fig 8 to be enough to remove a specific temporal bias from the evolution of the available satellite constellations.

**Changes**: no changes

**Comment 6:**

**Referee:** Significance of the trends being depending on the selected period, did you experiment trends on 25 or 30 years (30 years being the WMO and good pratice in climate)?

**Authors' response:** The issue is that there are several 30-year period possibilities since our SMOD dataset covers 38 years. Therefore we prefer to show the results for the longest possible period. However, 30-year trends are shown in Fig 11 and Fig 12 for two massifs, to illustrate the sensitivity of the results to the computation period.

**Changes**: no changes

**Comment 7:**

**Referee:** Why the absence of in situ stations in the Spanish Pyrenees is not more discussed?

**Authors' response:** There are four stations in the Spanish Pyrenees but indeed the majority of the stations is in France. We did not discuss this country-wise discrepancy because this dataset only served to evaluate the SMOD algorithm, not to compute the trends. In general we can emphasize in discussion that our results are more uncertain in Spain due to the lack of evaluation but also due to fewer SPOT images.

**Changes**: emphasis added in the discussion (that there were fewer SPOT images over Spain was already addressed)

**Comment 8:**

**Referee:** Why no percentage of pixel per altitude per massif is provide? This could have lead to more analysis on the differences in the trends per massif. In the same state of mind, why DAH wasn't more investigated?

**Authors' response:** We stratified the trend analysis by elevation range to allow the comparison of trends between different regions as done previously by e.g. Matiu et al. (2021). We can provide information on the hypsometry of every massif however we do not see how this can influence the interpretation of the results since we have made sure that each elevation band has a sufficient amount of data for the trend analysis (at least 1000 pixels).

**Changes**: We finally decided to not do provide a hypsometry and made no change related to this comment.

**Comment 9:**

**Referee:** In general, captions for figures are a little dry in the main article, whereas too many information (like explainations and comments) are present in the captions of figures in Annexes. For example, Fig. 2 could be more explicitly described (proportion of available data Pyrenees vs Alps; specificities for each mountain range regarding the number of observation as a fonction of altitude; why about all the southern part of Pyrenees doesn't have selected in situ data).

**Authors' response:** We thank the referee for this comment, we will elaborate the caption of Figure 2 as suggested.

**Changes**: We added more information in the captions

**Comment 11:**

**Referee:** Fig. 3: It could be refered at the end of the introduction of section 2.1, in order to give an overview of the satellite products considered in the study.

**Authors' response:** This figure cannot be easily shown in the introduction as the acronyms and dataset (Theia, SWHLX and DLR) are introduced later in the Data section.

**Changes**: no changes

**Comment 12:**

**Referee:** Fig. 5: Is Alpe d'Huez all the green square (if so, remove the texte from SPOT-Landsat figure and add it on the top figure) or does it refer to a specific location (if so, please use a symbol to locate Alpe d'Huez and put it on the three figures)?

**Authors' response:** We will remove the "Alpe d'Huez" textbox from the left panel of the figure.

**Changes**: "Alpe d'Huez" removed

**Comment 13:**

**Referee:** Fig. C5: Why the legend only includes few years, whereas more are presented in the figure?

**Authors' response:** There are 9 years (1987-1995), we agree that the legend is confusing, we will rework it.

**Changes**: all years are now in the legend

**Comment 14:**

**Referee:** Fig. C6: The color used fot NOBSmin=10 is almost invisible.

**Authors' response:** We agree. We will find a better color palette.

**Changes**: we changed the color

**Other comments :**

**Referee:**

L24: Is it «air temperature» or «near-surface air temperature»?

L49: Please add «in the European Alps» after Monteiro and Morin (2023).

L73: Please explicit the SPOT accronym here instead of L74.

L96: For more clarity between satellite data and providing centers, please change «Theia (Sentinel-2A&B and Landsat 8)» for «Theia L2-Snow Product (Sentinel-2A&B and Landsat 8), hereafter Theia»

L99: Please explicit «NIR».

L119: Please explicit «SCA».

L129: Please add «(Fig. 2)» at the end of the first sentence.

L144: Please change «SWHLX3» for «SWHLX».

L199: Please explicit NOBS.

L216: Please change «Mann–Kendall (MK)» for «MK».

L299: There is a missing space between «Ossau.» and «However».

L304: Please change «Pyrenees 12» for «Pyrenees (Fig. 12)».

L428: Please change «(Barrou Dumont et al., 2024b)» for «Barrou Dumont et al. (2024b)».

Fig.1: Adding the borders can be useful as you give information about the Pyrenees or the French Pyrenees. This figure can be presented in the in situ data section, instead of the introduction.

Fig.2: Please use the same legend as in Fig.1 for mountains ranges, and explicit that « o » symbol corresponds to stations. Are the barplots stacked or superposed?

Fig. 6: Left and right figures present the same information for Theia. Please remove the left figure, and put statistic information in a table or near the figure specifying which stats are for which dataset (Theia or Theia+SWHLX). Why the statistics for SWHLX alone are not presented?

Fig. C1,2,3: Please use a date spelling like «1 February 2023» (British English) or «February 1, 2023» (American English), to avoid confusion about the date.

Fig. C4: Please correct the caption with ΔSMOD instead of SMOD.

**Changes**: all the changes were applied

**Anonymous Referee #2**

**Comment 1:**

**Referee:** This study uses multi-source datasets to reveal snow melting day over the long time series. Employing various methods, it improves the spatial-temporal resolution of snow melting day to support the study results. Authors have made a comparison between SWH and landsat, but I am also worried about the data consistency between different landsat sensors, and between landsat and sentinel-2, especially under the case with a long study period and the mixture of all available data. So my suggestion is to add some data validation before using satellite datasets to calculate snow melting day.

Line 115, before mixing sentinel-2 and landsat-8 data from 2016 to 2023, I miss the consistency check between both datasets or sensor correction because they are from different sensors. Only if the data from different sensors are proven to be consistent, the trends calculated later will be meaningful.

**Authors' response:** We agree that data consistency is important. First of all we note that all sensors are multispectral imagers at 20-30m resolution, i.e. we do not blend high and low resolution imagery (like MODIS, AVHRR), or radar (like ERS or Sentinel-1). Secondly, we have evaluated the annual SMOD products that results from our multi-sensor time series and assessed their uncertainty in the paper. In the end, this is what matters for us since our objective is the recent evolution of the SMOD. We did not detect any trend in the mean SMOD error, but a larger spread in the beginning of the study period, that we attribute mainly to the lower revisit. In this paper we focused on the comparison between Landsat-DLR and SPOT SWH products because these products result from different retrieval algorithms (NDSI-based unsupervised classification vs. deep learning). Landsat products before 2018 (Landsat-DLR) were already evaluated by Hu et al. (2019a, 2019b). The performances of SPOT4 products were also previously evaluated by Barrou Dumont et al. (2024), and the spectral characteristics of the SPOT products are essentially near-identical outside of the addition of the SWIR band to SPOT4 and 5.  However, in response to the reviewer we have performed an additional analysis to evaluate the agreement between Landsat and Sentinel-2 products after 2016 as suggested.

We performed a comparison between eight pairs of snow products from Landsat 8 (L8) and Sentinel-2 (S2). These image pairs were acquired on the same day between November 2017 and June 2018, covering areas in the Alps and the Pyrenees.

**Changes**: An additional analysis of the agreement between landsat and Sentinel-2 was added in the appendix

**Comment 2:**

**Referee:** Line 110, when you combined the landsat data from different sensors over the long time series, how did you consider the sensor correction or data consistency between products from various sensors? In addition, can you give a brief description regarding the algorithm to produce level 2B product?

**Authors' response:** We retrieved the snow cover area from Level-2 Landsat Collection 2 Level-2 that are processed by USGS to provide consistent surface reflectance across different Landsat sensors and thus enable time series analysis (Köhler et al. 2022). The USGS applies radiometric calibration and atmospheric correction algorithms to Level-1 Landsat data. We can give a brief description of the Level-2B algorithm (snow detection) in a revised manuscript.

**Changes**: a brief description of the Level-2B algorithm was added in the data section

**Comment 3:**

**Referee:** Line 183, what is the basis to set the threshold of tree cover 50%?

**Authors' response:** We aimed to exclude dense forests where the snow detection is very uncertain with multispectral optical sensors such as Landsat and Sentinel-2 - without excluding sparse forests where the snow detection is still accurate (Muhuri et al. 2021, Barrou Dumont et al. 2021, Figure 3). We note that the TCD distribution across our study domain is bimodal with a mode near 0% and a mode near 100%. Therefore we expect the sensitivity to this value to be low.

**Changes**: no changes

**Comment 4:**

**Referee:** Check the figure 5. The left sub-figure is landsat only or landsat+spot?

**Authors' response:** There was indeed a mistake in the caption. The left panel is SPOT+Landsat.

**Changes**: caption corrected

**Comment 5:**

**Referee:** Fig. 9 and Fig. 10, how about mapping pixel-wise? One of advantages of this study is high spatial resolution.

**Authors' response:** We took advantage of the spatial resolution by stratifying the analysis in 300 m elevation bands up to 3600 m high. The high resolution allowed us to obtain a large number of pixels even at high elevations, which would not be possible with coarser resolution sensors. In addition, the aggregation by elevation and massif contributes to reduce the noise at the pixel level (random error) and thus to show more robust results. A map at 20 m resolution would have to be resampled to a much coarser resolution to fit in the article format (without resampling at 72 dpi, a 300 km wide domain such as the Pyrenees

measures approximately 5 m wide). This is why we choose to show only a small region at full resolution (Figure 5). Finally, the products are available at full resolution in a public repository.

**Changes**: no changes

**Comment 6:**

**Referee:** Line 322, I am not sure how tree cover density correlates to elevation? Maybe the changes in bias with elevation could be partly explained by tree cover density. How about dividing tree cover density into several classes? Then analyze something, such as the bias and trend.

**Authors' response:** Following this suggestion we have computed the correlation of TCD and elevation from all the pixels corresponding to elevations > 1200m and TCD > 0 and TCD < 50. We obtained a correlation of -0.01 between TCD and elevation (therefore no correlation between the two). We will add this point to the Discussion.

**Changes**: TCD/elevation correlation added to the discussion

**Additional changes**

- Figures C7-C10 have been moved to a supplement document
- The generation of DLR-Landsat products over the pyrenees was moved from the method to the data section
- corrected in the abstract the number of stations after filtering for coordinates with low precision.
- added the author's affiliation to Magellium